# Multiple patterns of persistent inward currents with multiple types of repetitive firings in medullary serotonergic neurons of mice: An experimental and modeling study

Yi Cheng[1], Xingyu Wang[2], Qiang Zhang[3], Renkai Ge[4], Mei Zhou[2], Yue Dai[1,2*]

**1** Key Lab of Adolescent Health Assessment and Exercise Intervention of Ministry of Education, College of Physical Education and Health Care, East China Normal University, Shanghai, China, **2** Shanghai Key Laboratory of Multidimensional Information Processing, School of Communication and Electronic Engineering, East China Normal University, Shanghai, China, **3** School of Electrical and Information Engineering, Jiangsu University of Science and Technology (Zhangjiagang Campus), Zhangjiagang, China, **4** School of Physical Education and Health Care, East China Jiaotong University, Nanchang, China,

* ydai@tyxx.ecnu.edu.cn

## Abstract

Persistent inward currents (PICs) play a crucial role in regulating neuronal excitability. These currents are composed of calcium (CaL) and sodium (NaP) components in vertebrate spinal neurons. Recent studies have reported that PICs are expressed in serotonergic neurons (5-HT) in medulla of mice. Multiple patterns of PICs were identified in 5-HT neurons, corresponding to a range of distinct repetitive firing types. The mechanisms underlying formation of these PIC patterns and firing types remain unknown. Using combined modeling and experimental approaches we explored the ionic mechanisms responsible for the PIC patterns and firing types. The whole cell patch clamp recordings were performed on the medullary 5-HT neurons of postnatal day 3–6 mice. A 5-HT neuron model was built based on the membrane properties of the 5-HT neurons and kinetics of voltage-gated channels. Results from physiological experiments and modeling simulations included: (1) PICs in 5-HT neurons were classified into six patterns based on their current trajectory induced by bi-ramp voltage, while repetitive firings were categorized into three types according to their response to bi-ramp current. Modulation of PICs conductance and kinetics altered the PIC patterns and firing types. (2) NaP conductance contributed to amplitude of PICs, whereas the slow inactivation gate ($S_{gate}$) of NaP regulated the PIC patterns and firing types. Increasing $S_{gate}$ changed trajectory of PICs from counterclockwise to clockwise and firing types from asymmetrical to symmetric types induced by bi-ramp current. (3) CaL conductance dominated the amplitude of PICs, while CaL kinetics (half-activation voltage and slope) determined inactivation of PICs and prolongation of repetitive firing. (4) The novel finding was that distribution of CaL in distal dendrites modulated the PIC patterns and firing types. This study provides insights into the ionic mechanisms underlying generation of multiple PIC patterns and firing types in 5-HT neurons.

**Data availability statement:** All electro-physiological experimental data and model code are available at: https://github.com/Cheng-ECNU/5-HT-model-PIC

**Funding:** This research was funded by the National Natural Science Foundation of China Grant Number 32471187 and 32171129 for YD and by China Postdoctoral Science Foundation Grant Number 2023M731112 for YC. The funders had no role in study design, data collection and analysis, decision to publish, or preparation of the manuscript.

**Competing interests:** The authors have declared that no competing interests exist.

## Author summary

Persistent inward currents (PICs) of medullary serotonergic (5-HT) neurons play an important role in regulating neuronal excitability and facilitating locomotion. However, the ionic basis responsible for this process remains unclear. Using modeling and experimental approaches we investigated the mechanisms underlying the patterns of PICs and types of repetitive firing in the 5-HT neurons of mice. Six patterns of PICs with three types of repetitive firing were discovered in medullary 5-HT neurons of postnatal day 3–6 mice. The sodium component of PICs (NaP) predominantly contributes to the threshold of PICs. Specifically, the slow inactivation of NaP played an essential role in regulating the PIC patterns and firing types. On the other hand, however, the calcium component of PICs (CaL) dominated the amplitudes of PICs with major contributions to the inactivation of PICs and prolongation of firings. Both NaP and CaL played essential in roles in regulating neuronal excitability but targeted different biophysical parameters. More importantly, for the first time we demonstrated that distal distribution of CaL in dendrites determined the patterns of PICs and types of repetitive firings in medullary 5-HT neurons of mice. This study unveiled the putative, PIC-mediated mechanisms underlying regulation of neuronal excitability and initiation of locomotion in mice.

## Introduction

Persistent inward currents (PICs) are depolarizing currents generated by slow inactivation voltage-gated sodium (NaP) channels and L-type calcium (CaL) channels that have been found in many types of neurons in vertebrates [1–4]. PICs are crucial for various physiological processes in neurons, including the generation and modulation of action potentials, amplification of synaptic current, and regulation of neuronal excitability [5–8]. PICs have regulatory effects on various functional activities such as memory, respiration, and locomotion [9–11]. PICs play an essential role in generating rhythmic motor patterns of locomotion through modulating the excitability and firing properties of neurons constituting the central pattern generator (CPG) [12–14]. In recent study, we find that PICs also play a crucial role in modulating medulla serotonergic (5-HT) neurons which are involved in initiating and controlling locomotion [15–21].

Originating from medulla, the descending serotonergic pathway exerts profound influences on locomotion by modulating spinal cord CPG networks, controlling motoneuron excitability, and generating the rhythmic activities of locomotion [18,22,23]. In our recent studies we have reported some novel results about functional roles of PICs in modulating 5-HT neurons [20,21,24]. PICs in medullary 5-HT neurons can be classified as an ascending PIC (a-PIC) evoked in the rising phase of the ramp voltage and a descending PIC (d-PIC) generated in the falling phase of the ramp [20]. The trajectory of the PIC current shows the evolution of the current vs voltage. This trajectory can be counterclockwise if a-PIC is larger than d-PIC, otherwise a clockwise trajectory of PIC forms [3]. Clockwise and counterclockwise trajectory are traditional methods used to characterize PICs [1,3,20]. It has been shown in previous studies that the clockwise and counterclockwise trajectory of the PICs determines the firing frequency of neurons under bi-ramp current stimulation [1,6,25,26]. The firing frequency in the rising phase of the bi-ramp current is generally larger than that in the falling phase if a-PIC > d-PIC. Conversely, the firing frequency in the rising phase is less than that in the falling phase if a-PIC < d-PIC.

Based on the PIC definitions, we discovered that PICs in 5-HT neurons exhibited multiple patterns which regulated the neuronal excitability and firing properties [21,24,27]. We also found three types of repetitive firings in 5-HT neurons induced by bi-ramp currents [21]. Furthermore, we discovered that the mechanism underlying staircase PICs in medullary 5-HT neurons, a special shape of PICs composed of two staircase-like inward currents, was induced by sequential activation of NaP and CaL components of PICs [21]. However, the mechanisms underlying generation of multiple patterns of PICs and different types of repetitive firing are not well understood, yet. Especially, the relationships between the patterns of the PICs and types of repetitive firing remain unknown.

The purpose of study is to explore the correlation between PIC patterns and discharge types in medullary 5-HT neurons of neonatal mice. To this end we employed electrophysiological experiments combined with modeling approach to investigate the contributions of NaP, CaL and neuronal morphology to PIC patterns and discharge types. Simulation results indicated that NaP mainly contributed to the counterclockwise trajectory of PICs with premature neuronal discharge, while CaL was responsible for the clockwise trajectory of the PICs with prolongation of neuronal firing. More importantly, we discovered that distal distribution of CaL in dendrites of 5-HT neurons produced a delayed inactivation of PICs which mediated postponed repetitive firing.

## Results

### Classification of PIC patterns and firing types

Using transverse sections of the medulla from transgenic ePet-EYFP mice aged 3–6 days, we localized 5-HT neurons expressing fluorescent proteins in the parapyramidal region (PPR) and midline raphe nuclei (MRN) of the medulla (Fig 1A). A slow triangular bi-ramp voltage was used to induce the PICs in the 5-HT neurons (Fig 1B). Based on the ascending and descending phases of the bi-ramp, the PICs were defined as a-PIC and d-PIC, respectively (see Methods). The activation ($V_{onset}$) and inactivation ($V_{offset}$) voltage thresholds were measured. The difference between them ($\Delta V = V_{offset} - V_{onset}$) was used to categorize the PIC patterns (see Methods). Similarly, a slow triangular current ramp was injected into the 5-HT neurons to evoke repetitive firing (Fig 1C). The onset ($I_{onset}$) and offset ($I_{offset}$) current thresholds were measured. The difference between them ($\Delta I = I_{offset} - I_{onset}$) was used to classify the firing types (see Methods). It is noted that the PIC patterns are always overlapped with firing types in physiological experiments, and it is technically difficult to establish a clear relationship between $\Delta V$ and $\Delta I$. Therefore, we apply modeling approach to study this issue.

A single-cell model composed of axon, initial segment, soma, and dendrite compartments was built in this study (Fig 1D and Methods), based on the membrane properties of 5-HT neurons (Table 1). Six active conductances were included in the model (Tables 2 and 3). The membrane properties of 5-HT neurons and the model were summarized in Table 1. This model was used to explore the correlation between PIC patterns and firing types of medullary 5-HT neurons in mice. In order to identify the individual contribution of NaP or CaL to generation of PIC patterns and firing types, we blocked one channel (set conductance to zero) to test the other one in the simulations.

### Multiple patterns of PICs and firing types in 5-HT neurons

Similar to the results from our recent studies, PICs were widely expressed in medullary 5-HT neurons [20,24]. Six patterns of PICs were found in the 47 neurons recorded (Fig 2A). The first pattern of PICs had a-PIC only with a $V_{onset}$ and $\Delta V > 0$. This pattern presented in a counterclockwise trajectory (a-PIC > d-PIC) of PIC (18/47, Fig 2A1). The second pattern had

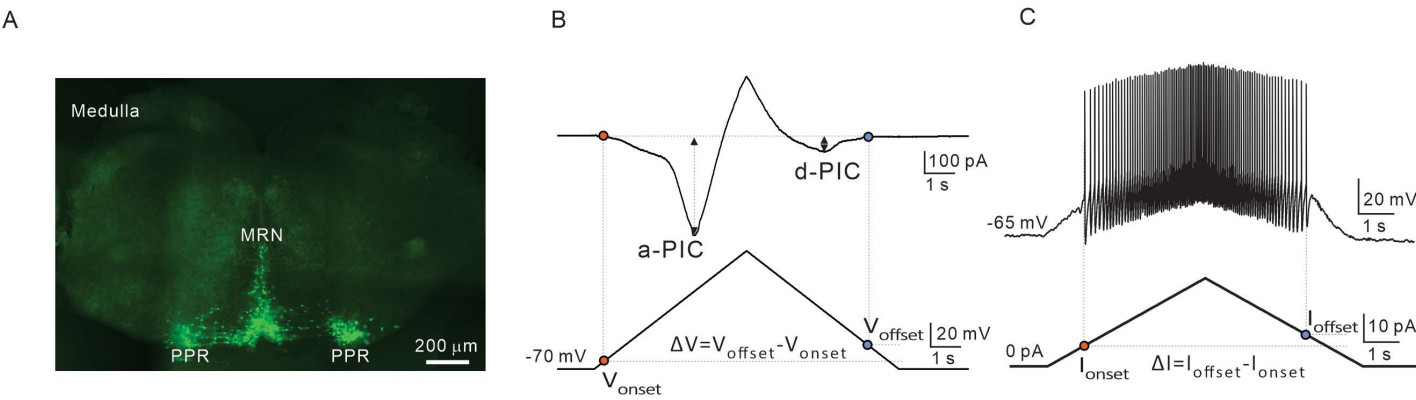

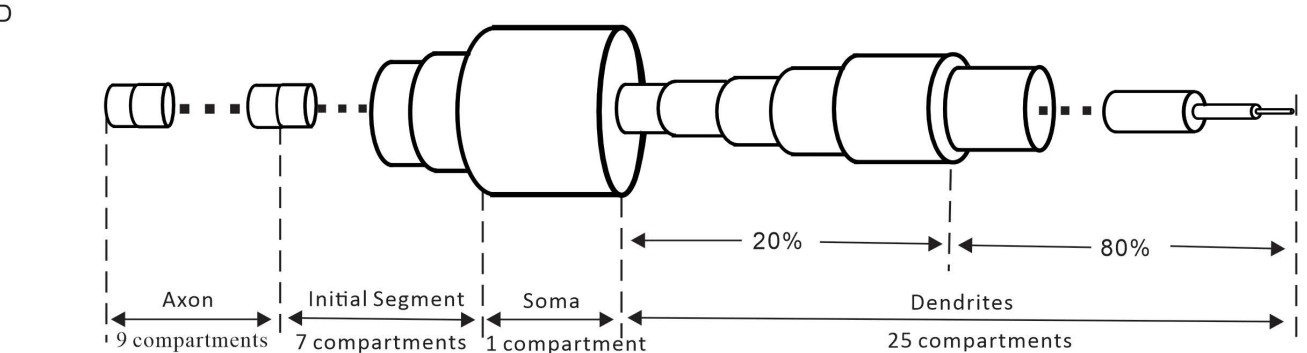

**Fig 1. PICs and repetitive firing in 5-HT neurons with the 5-HT neuron model. A.** A medullary slice of ePet-EYFP mice shows the 5-HT neurons located in the parapyramidal region (PPR) and midline raphe nuclei (MRN). **B.** Measurement of biophysical parameters of PICs ($V_{onset}$, $V_{offset}$, a-PIC, d-PIC and $\Delta V$) by bi-ramp-voltage. **C.** Measurement of biophysical parameters ($I_{onset}$, $I_{offset}$, and $\Delta I$) from repetitive firing induced by bi-ramp current. **D.** A 5-HT neuron model with 4 section lists (axon, initial segment, soma, and dendrite) was built with membrane properties of medulla 5-HT neurons of mice. This panel D was hand-drawn by us.

**Table 1. Membrane properties of 5-HT neurons and model.**

| Property | Physiological data (n=16) | Model |
|---|---|---|
| $E_m$ (mV) | −56.79±3.54 | −62 |
| $I_{th}$ (pA) | 15±7 | 10 |
| $V_{th}$ (mV) | −36.3±3.5 | −38.9 |
| AP height (mV) | 56.66±7.177 | 62 |
| AP 1/2 width (ms) | 1.983±0.5381 | 1.1 |
| AHP amplitude (mV) | 22.42±4.98 | 19.12 |
| AHP 1/2 width (ms) | 282±122.83 | 352.4 |
| $R_{in}$ (M$\Omega$) | 826.413±273.085 | 776 |

$E_m$: resting membrane potential, $I_{th}$: current threshold, $V_{th}$: voltage threshold, AP: action potential, AHP: afterhyperpolarization, $R_{in}$: input resistance. Physiological data are represented as means ± SD.

both a-PIC and d-PIC with a-PIC amplitude larger than the d-PIC, and $V_{onset} < V_{offset}$, $\Delta V > 0$. This pattern exhibited a counterclockwise trajectory (14/47, Fig 2A2). The third pattern had both a-PIC and d-PIC with a-PIC amplitude larger than the d-PIC and $V_{onset} = V_{offset}$, $\Delta V = 0$.

**Table 2. Structure of 5-HT neuron model and cable parameters.**

| Neuron | diameter | length | $R_M$ | $R_A$ | $C_M$ |
|---|---|---|---|---|---|
| Compartment | (μm) | (μm) | (Ωcm²) | (Ωcm) | (μF/cm²) |
| Axon | 1.8 | 20 | 50000 | 70 | 1 |
| Initial segment (IS) | 1.8~8.2 | 10 | 50000 | 70 | 1 |
| Soma | 12 | 20 | 16666 | 70 | 1 |
| Dendrite | 1.4~5.4~0 | 600 | 34483 | 70 | 1 |

**Table 3. Distribution and density of active conductances.**

| Distribution | $G_{max}$ (mS/cm²) |
|---|---|
| IS/Axon conductances | |
| NaT | 139 |
| NaP | 0.158 |
| Kdr | 40 |
| Kleak | 0.001 |
| Soma conductances | |
| NaT | 30 |
| NaP | 0.1 |
| Kdr | 10 |
| KCa | 0.18 |
| Kleak | 0.04 |
| Dendrite conductances | |
| NaT (0:0.04) | 4.4 |
| NaT (0.04:1) | 0.0075 |
| NaP (0:0.04) | 0.0044 |
| NaP (0.04:1) | 0.00015 |
| Kdr (0:0.04) | 1 |
| Kdr (0.04:1) | 0.00033 |
| KCa (0.32:0.56) | 0.04 |
| CaL (0.32:0.56) | 0.03 |
| Kleak | 0.01 |

IS and Axon have the same conductance densities, including transient sodium conductance (NaT), persistent sodium conductance (NaP), and delayed rectifier potassium conductance (Kdr). Soma contains four conductance: NaT, NaP, Kdr, and calcium-activated potassium conductance (KCa). More NaT, NaP, and Kdr conductance densities on most proximal segment of dendrite where from 0 to 0.04 away from the soma, and very few conductance densities are located at distal dendrite. Dendrite KCa and CaL distributed at the interval 0.32 to 0.56 from the soma.

This pattern showed a counterclockwise trajectory (5/47, Fig 2A3). The fourth pattern had d-PIC only with a $V_{offset}$ and $\Delta V < 0$. This pattern produced a clockwise trajectory of PIC (2/47, Fig 2A4). The fifth pattern had both a-PIC and d-PIC with a-PIC amplitude smaller than the d-PIC and $V_{onset} > V_{offset}$, $\Delta V < 0$. This pattern displayed a clockwise trajectory (5/47, Fig 2A5). The sixth pattern expressed both a-PIC and d-PIC with a-PIC larger than the d-PIC and $V_{onset} > V_{offset}$, $\Delta V < 0$. A counterclockwise trajectory of PIC was shown in this pattern (3/47, Fig 2A6). Of 47 neurons, the proportions of the six types of PICs were 38%, 30%, 11%, 4%, 11%, and 6%, respectively. In this study, we focused on the difference of PIC $V_{onset}$ and $V_{offset}$, and divided PICs into three categories: $\Delta V > 0$, $\Delta V = 0$, and $\Delta V < 0$. Statistical results from 47 recorded 5-HT neurons showed that 68%,11% and 21% of the 5-HT neurons displayed the $\Delta V > 0$, $\Delta V = 0$, and $\Delta V < 0$, respectively (Fig 2C).

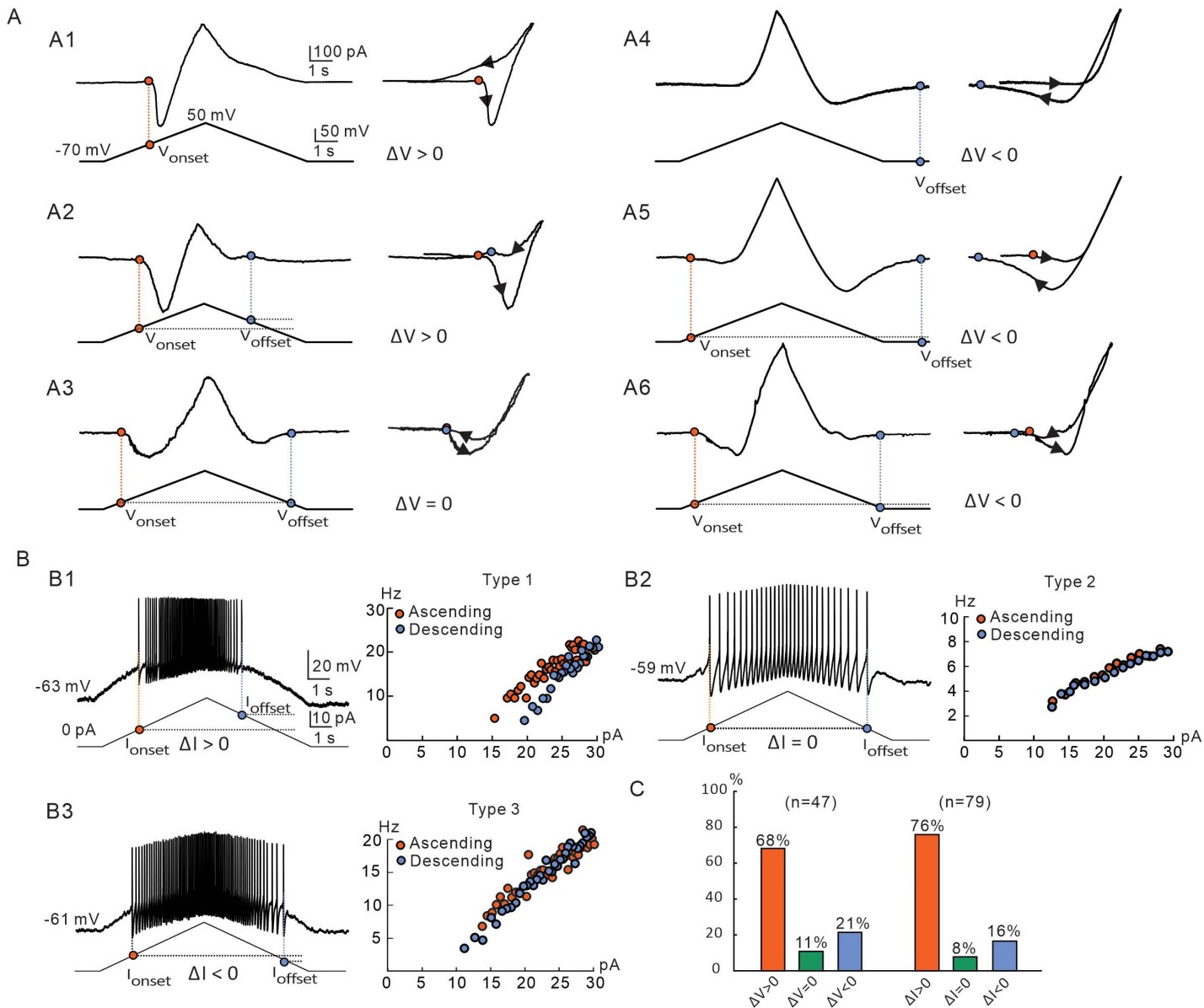

**Fig 2. Multiple patterns of PICs and multiple types of firing in 5-HT neurons. A.** Six patterns of PICs in 5-HT neurons. (A1) The PICs had only a-PIC with counterclockwise trajectory and ΔV > 0, current trajectory recorded by bi-ramp voltage from −70 mV to 50 mV (bottom). (A2) PICs displayed a-PIC > d-PIC with counterclockwise trajectory of PICs and ΔV > 0. (A3) PICs exhibited a-PIC > d-PIC with counterclockwise trajectory and ΔV = 0. (A4) PICs had only d-PIC with clockwise trajectory and ΔV < 0. (A5) PICs showed d-PIC > a-PIC with clockwise trajectory and ΔV < 0. (A6) PICs displayed a-PIC > d-PIC with counterclockwise trajectory and ΔV<0. The orange circles and dashed line indicated the $V_{onset}$ corresponding to PIC activation, while the blue circles and dashed line represented the $V_{offset}$ corresponding to PIC termination. **B.** Three types of repetitive firing induced by bi-ramp current in 5-HT neurons. (B1-B3) Repetitive firing with ΔI > 0 (Type 1), ΔI = 0 (Type 2), and ΔI < 0 (Type 3), were recoded, respectively. The orange and blue circles on the bi-ramp currents (left panels of B1-B3) represented the onset ($I_{onset}$) and offset ($I_{offset}$) currents which generated the first and last action potentials, respectively, during the repetitive firing. The frequency-current relationships of the instantaneous firing were established in the right panels of B1-B3. Orange and blue circles represented the ascending (orange) and descending (blue) frequencies, respective. **C.** Proportions of three types of PIC trajectories (ΔV > 0, ΔV = 0, ΔV < 0, n=47) and repetitive firings (ΔI > 0, ΔI = 0, ΔI < 0, n=79).

Based on previous studies, PICs contributes to neuronal repetitive firing in response to bi-ramp current stimulation [20,21,27,28]. Three types of firing were found in 79 recorded 5-HT neurons (Fig 2B). The first type (type 1) exhibited the ΔI >0 (Fig 2B1), the second type (type 2) ΔI = 0 (Fig 2B2), and the third type (type 3) ΔI <0 (Fig 2B3). Statistical results showed

that 76% of 5-HT neurons (60/79) exhibited the first type of firing ($\Delta I > 0$), 8% of the neurons (6/79) the second type ($\Delta I = 0$), and 16% (13/79) the third type ($\Delta I < 0$) (Fig 2C). We also calculated the instantaneous firing frequency of 5-HT neurons during ascending (orange circles) and descending (blue circles) phases of the bi-ramp current (Fig 2B1, 2B2 and 2B3, right). These results showed that there were multiple patterns of PICs and three types of repetitive firing in medullary 5-HT neurons.

## Contributions of NaP and CaL to PIC patterns and firing types

Many studies have demonstrated the essential role of PICs in regulating excitability and recruitment of spinal motoneurons [13,29]. However, the functional roles of NaP and CaL in medullary 5-HT neurons remains unclear. In this study, we used modeling approach (Fig 1D) to explore the regulatory effect of NaP and CaL on the firing properties and excitabilities of 5-HT neurons. The data presented in the following were all simulated from the model of 5-HT neurons (see Methods).

### NaP contribution

NaP is an important component of PICs in 5-HT neurons [3,4,30]. In previous research we found that the low threshold of NaP constituted the primary component of the first step of staircase PICs [21] and played a crucial role in regulating the rheobase of 5-HT neurons [20]. In the following simulation, we investigated the effect of modulating NaP on PIC patterns and neuronal firing types.

**The modulation of NaP conductance.** We first investigated the modulatory effect of NaP maximal conductance (gNaP) on PIC patterns and firing properties of 5-HT neurons. In this case, CaL was blocked in the model (gCaL= 0). In our recent study, we found that chronic exercise increased the amplitude of NaP in medullary 5-HT neurons [27], suggesting that NaP in 5-HT neurons was regulatable during physiological activities. Simulation results showed that increasing the gNaP by 300% increased the amplitudes of both a-PIC and d-PIC without altering the counterclockwise trajectories of PICs (Fig 3A). In this case, an increased firing frequency was observed (Fig 3B). We analyzed the changes in $V_{onset}$ and $V_{offset}$, as well as $I_{onset}$ and $I_{offest}$, as gNaP was increased sequentially (Fig 3C and 3D). The $V_{onset}$ and $V_{offset}$ showed a linearly decreasing with increasing the gNaP (Fig 3C), and the same trend was shown in $I_{onset}$ and $I_{offset}$ (Fig 3D). Both $V_{osnet}$ and $I_{onset}$ were smaller than $V_{offset}$ and $I_{offset}$, respectively, suggesting that increasing the gNaP did not substantially alter the $\Delta V$ and $\Delta I$. Previous studies have indicated that the trajectory direction of PICs after folding along the midline depends on the amplitude differential between a-PIC and d-PIC [3]. Therefore, we further looked at the modulatory effect of increasing gNaP on a-PIC and d-PIC amplitudes (Fig 3E and 3F). Simulation results showed that a linear increase in the amplitudes of a-PIC and d-PIC was observed in this case (Fig 3E). However, the ratio of amplitude (d-PIC/a-PIC) did not change (Fig 3F). In summary, increasing the gNaP did not significantly alter the counterclockwise trajectory of PICs and firing property, suggesting NaP played a little role in regulating PIC patterns and firing types.

**The modulation of NaP kinetics.** In addition to the effects of modulating gNaP on the 5-HT neurons, we also explored the impact of NaP kinetics on 5-HT neurons. Simulation results showed that increasing the slope ($V_S$) reduced curve steep of the NaP activation (Fig 4A), which altered the onset and offset voltages of the PICs (Fig 4B). We mimicked the activation rate of NaP channels by altering the $V_S$ and then explored the output of the neuron with blockage of CaL (gCaL=0) during bi-ramp voltage-clamp and bi-ramp current-clamp recordings (Fig 4B and 4C). Increasing $V_S$ hyperpolarized $V_{onset}$ and depolarized $V_{offset}$ (Fig 4B

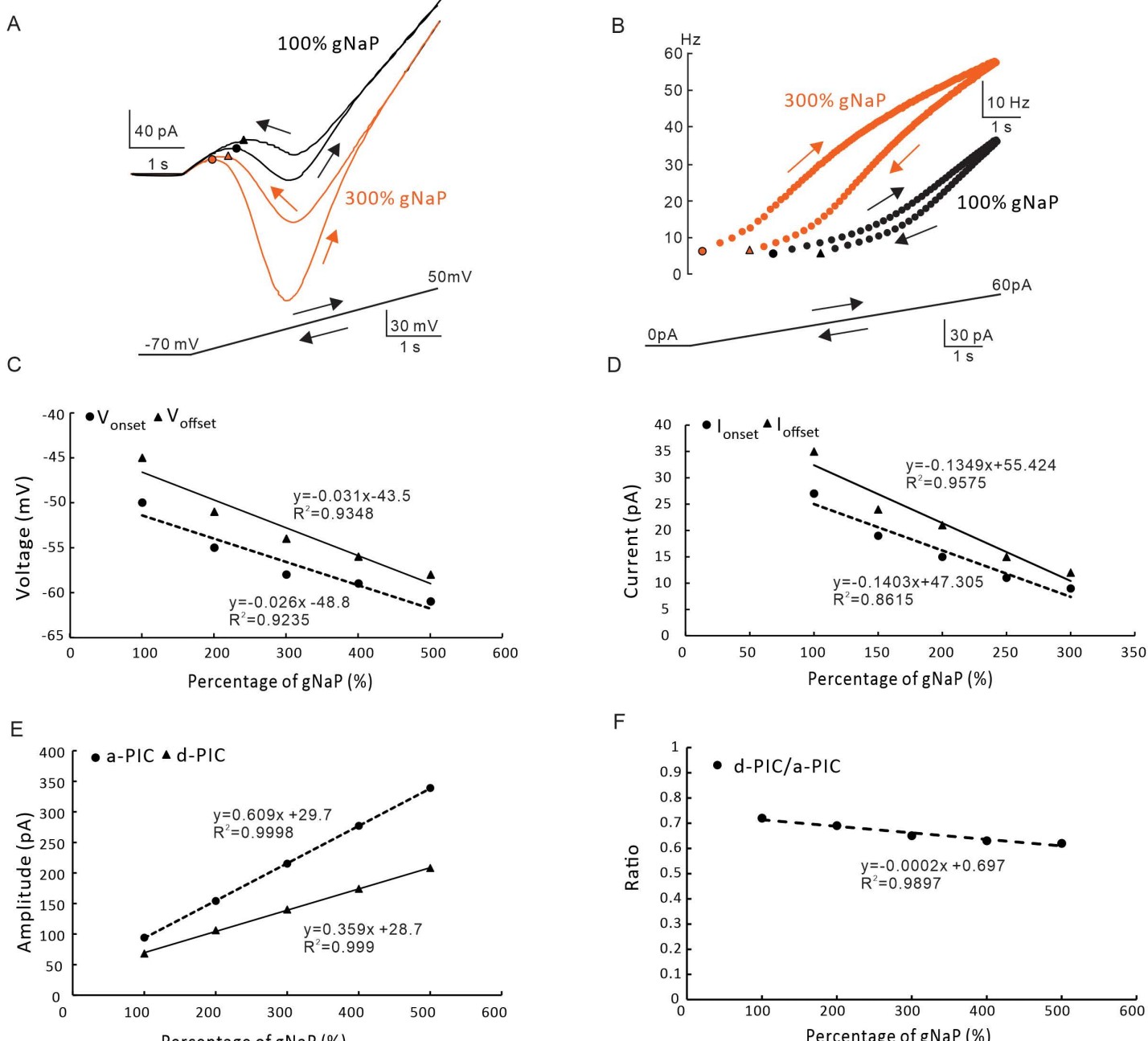

**Fig 3. The effect of NaP maximal conductance on PIC patterns and firing types. A.** PICs were induced by bi-ramps voltage (bottom) in 100% (black, control) and 300% (orange) of gNaP **B.** Instantaneous firing frequency produced by 100% (black, control) and 300% (orange) of gNaP. **C.** The relationship between the $V_{onset}$ & $V_{offset}$ and percentage of gNaP. Increasing the gNaP hyperpolarized the $V_{onset}$ and $V_{offset}$ with almost unchanged $\Delta V$. **D.** The relationship between the $I_{onset}$ & $I_{offset}$ and percentage of gNaP. Increasing the gNaP reduced the $I_{onset}$ and $I_{offset}$ with a little change in $\Delta I$. **E.** The relationship between the amplitude of a-PIC & d-PIC and percentage of gNaP. Increasing the gNaP increased amplitudes of a-PIC and d-PIC. **F.** The relationship between the gNaP and ratio of d-PIC/a-PIC. Increasing the gNaP slightly reduced ratio of d-PIC/a-PIC amplitudes. The closed circle symbols stand for $V_{onset}$, $I_{onset}$ and a-PIC, and the closed triangular symbols represent $V_{offset}$, $I_{offset}$ and d-PIC.

and 4D), leading to an increased $\Delta V$. We also analyzed the effect of increasing the $V_S$ on the firing property (Fig 4C and 4E). The results indicated that increasing $V_S$ led to a decrease in $I_{onset}$ and $I_{offset}$, ultimately resulting in an increased $\Delta I$. The amplitude of a-PIC decreased with the increase of $V_S$. Similarly, the trend of the amplitude of d-PIC was the same as a-PIC, but

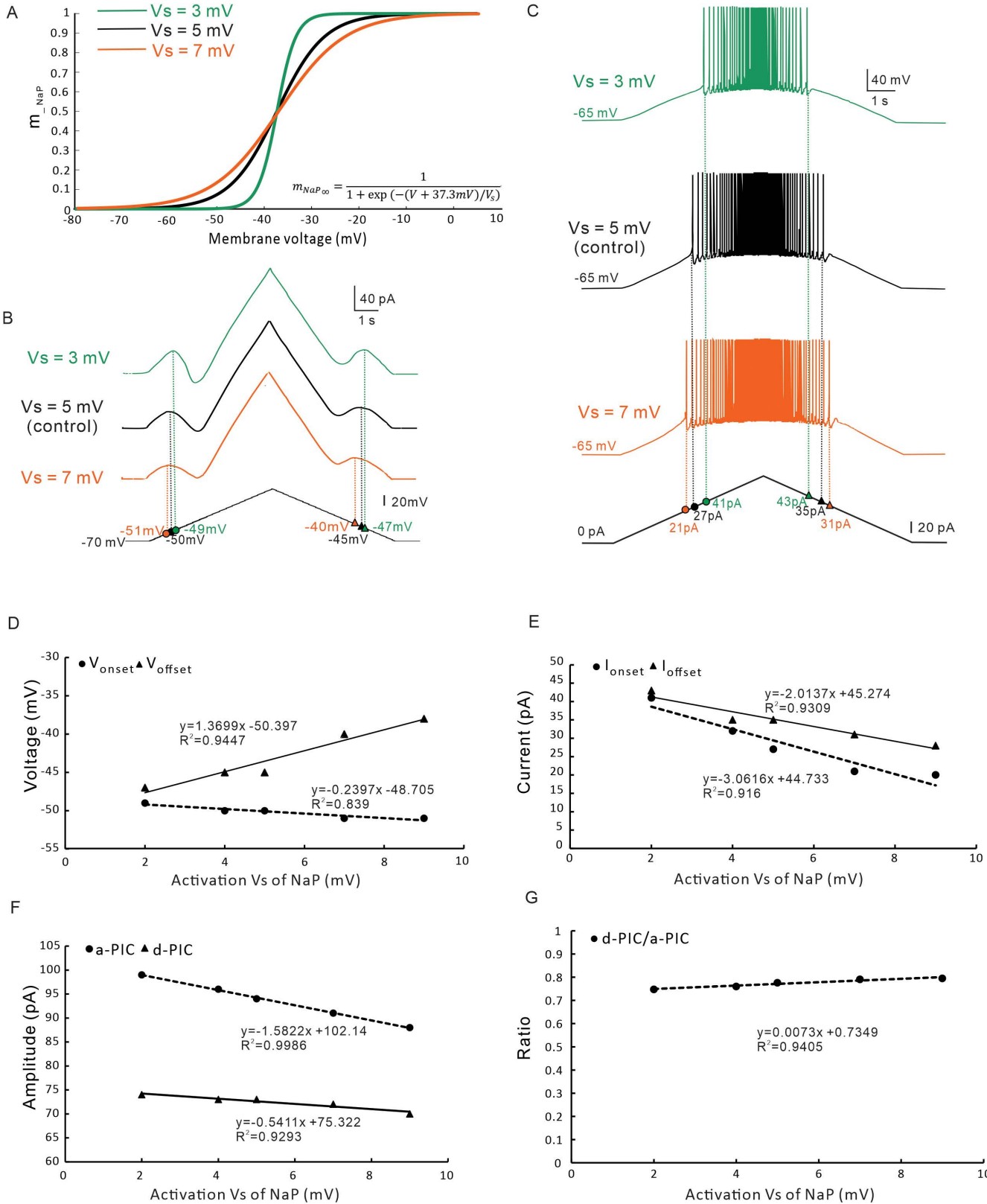

**Fig 4. The effect of kinetics $V_S$ of NaP on PIC patterns and firing types. A.** The steady-state of gating activation ($m_{NaP}$) curve of NaP was established when $V_S$ was set to 3 (green), 5 (black, control), and 7 mV (orange), respectively. **B.** PICs were induced with bi-ramp voltage when $V_S$ was set to 3 (green),

5 (black, control), and 7 mV (orange), respectively. **C.** Repetitive firings evoked with the bi-ramp current when $V_S$ was set to 3 (green), 5 (black, control), and 7 mV (orange). **D.** The relationship between the $V_{onset}$ & $V_{offset}$ and $V_S$ of NaP. Increasing the $V_S$ hyperpolarized the $V_{onset}$ and depolarized the $V_{offset}$ with an increased ΔV. **E.** The relationship between the $I_{onset}$ & $I_{offset}$ and $V_S$ of NaP. Increasing the $V_S$ of NaP reduced the $I_{onset}$ and $I_{offset}$ with an increased ΔI. **F.** The relationship between the amplitude of a-PIC & d-PIC and $V_S$ of NaP. Increasing the $V_S$ decreased amplitudes of a-PIC and d-PIC. **G.** The relationship between the ratio of d-PIC/a-PIC and $V_S$ of NaP. Increasing the $V_S$ of NaP did not substantially change the ratio of d-PIC/a-PIC amplitudes. The closed circle symbols stand for $V_{onset}$, $I_{onset}$ and a-PIC, and the closed triangular symbols represent $V_{offset}$, $I_{offset}$ and d-PIC.

the rate of change was lower than that of a-PIC (Fig 4F). Fig 4G demonstrated that the ratio d-PIC amplitude to a-PIC amplitude slightly increased as $V_S$ increased. These results indicated that increasing the $V_S$ of NaP had little effect on the counterclockwise trajectory (Fig 4F and 4G) of the PICs and the ΔV (Fig 4D), nor did it change the firing type of the neuron (Fig 4E). These results further suggested that $V_S$ of NaP channels had little impact on the patterns of PICs and firing types of 5-HT neurons.

Previous study has reported that the slow inactivation kinetics of NaP channels play a role in regulating neuronal firing [31]. Here, we explored the role of slow inactivation kinetics ($S_{gate}$) of NaP on PIC patterns and firing types. The $S_{gate}$ referred to the probability of NaP inactivation, ranging from 0 to 1 (see Methods). Fig 5 illustrated the effect of altering the slow inactivation variable $S_{gate}$ on the output of the neuron with blockage of CaL (gCaL = 0). As the $S_{gate}$ increased, the $V_{offset}$ and $I_{offset}$ decreased, and the $V_{onset}$ and $I_{onset}$ remained almost unchanged (Fig 5A and 5B). We systemically analyzed the changes in the $V_{onset}$, $V_{offset}$, $I_{onset}$ and $I_{offset}$ with increasing $S_{gate}$ from 0 to 1(Fig 5C and 5D). Simulation results showed that increasing $S_{gate}$ changed ΔV from positive (ΔV > 0) to negative (ΔV < 0), indicating an increased activation time of the PICs (Fig 5C). Increasing $S_{gate}$ also resulted in ΔI approaching to 0, indicating that neuronal firing was prolonged (Fig 5D). These results implicated that the slow inactivation of NaP mainly contributed to the offset voltage and offset current with little effect on onset voltage and onset current. It also suggested that slow inactivation gate was one of the main factors producing the delayed-inactivation of PICs. Simulation results also showed that amplitudes of a-PIC and d-PIC both increased with increasing $S_{gate}$ (Fig 5E). Furthermore, Fig 5F showed that the ratio of PIC amplitude (d-PIC/a-PIC) increased linearly with increasing $S_{gate}$. It is noted that the amplitude of a-PIC equaled to d-PIC when the gNaP did not inactivate ($S_{gate}$=1). The above results suggested that the $S_{gate}$ of NaP contributed to regulation of counterclockwise trajectory of PICs. Increasing $S_{gate}$ prolonged the duration of firing in descending phase of bi-ramp current thus increasing the excitability of neuron.

## CaL contribution

CaL is main component of PICs, which play a major role in generating plateau potentials, amplifying synaptic currents and promoting neuronal firing [32–33]. In a recent study, we found that CaL played an important role in prolonging firing of 5-HT neurons [20]. In the following study, we focused on the effects of modulating the maximum conductance, gating kinetics, and distribution of CaL on the PIC patterns and firing types of 5-HT neurons.

**Modulation of CaL conductance.** The effect of modulating CaL conductance (gCaL) on the neuron was studied with NaP conductance set to 0 in the model. After blocking the NaP, the CaL appeared in a clockwise trajectory (Fig 6A). Increasing the gCaL by 300% increased the PIC amplitude but did not change the clockwise trajectory of the PICs, suggesting that CaL should be a main factor determining the amplitude of PICs in 5-HT neurons (Fig 6A). This result was consistent with previous report in rat hypoglossal motoneurons [34–35]. We further analyzed the effect of modulating gCaL on the firing

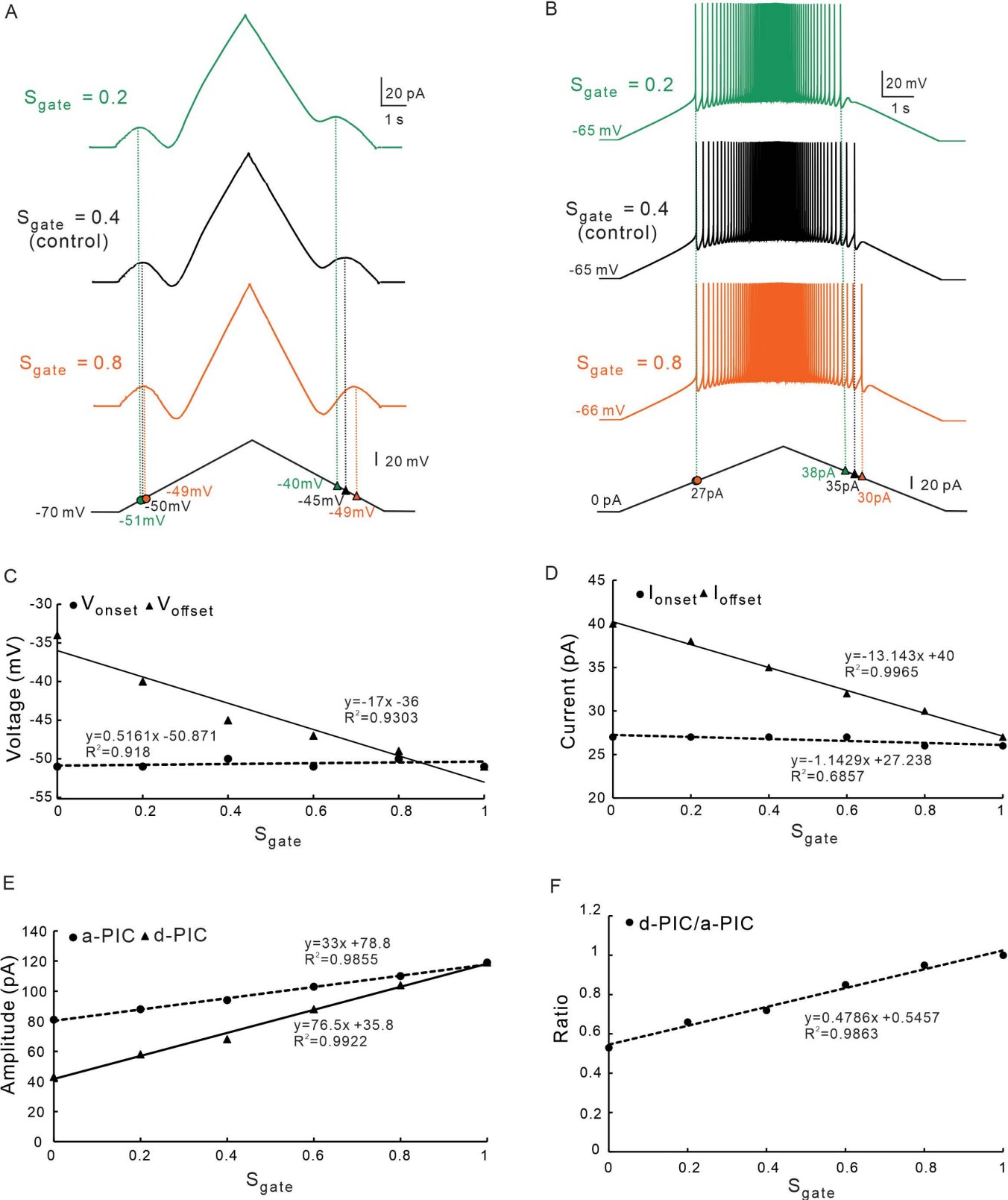

**Fig 5. The effect of slow inactivation variable $S_{gate}$ of NaP on PIC patterns and firing types. A.** PICs were induced with bi-ramp voltage when $S_{gate}$ was set to 0.2 (green), 0.4 (black, control), and 0.8 (orange), respectively. **B.** Repetitive firings induced by current clamp when $S_{gate}$ was set to 0.2 (green), 0.4 (black, control), and 0.8 (orange). **C.** The relationship between the $V_{onset}$ & $V_{offset}$ and $S_{gate}$ of NaP. Increasing the $S_{gate}$ of NaP did not change the $V_{onset}$ but hyperpolarized

the $V_{offset}$ with a decrease in $\Delta V$. **D.** The relationship between the $I_{onset}$ & $I_{offset}$ and $S_{gate}$ of NaP. Increasing the $S_{gate}$ did not change the $I_{onset}$ but reduced the $I_{offset}$ with a decrease in $\Delta I$. **E.** The relationship between the amplitudes of a-PIC & d-PIC and $S_{gate}$ of NaP. Increasing the $S_{gate}$ increased amplitudes of a-PIC and d-PIC. **F.** The relationship between the ratio of d-PIC/a-PIC and $S_{gate}$ of NaP. Increasing the $S_{gate}$ of NaP increased the ratio of d-PIC/a-PIC amplitudes. The closed circle symbols stand for $V_{onset}$, $I_{onset}$ and a-PIC, and the closed triangular symbols represent $V_{offset}$, $I_{offset}$ and d-PIC.

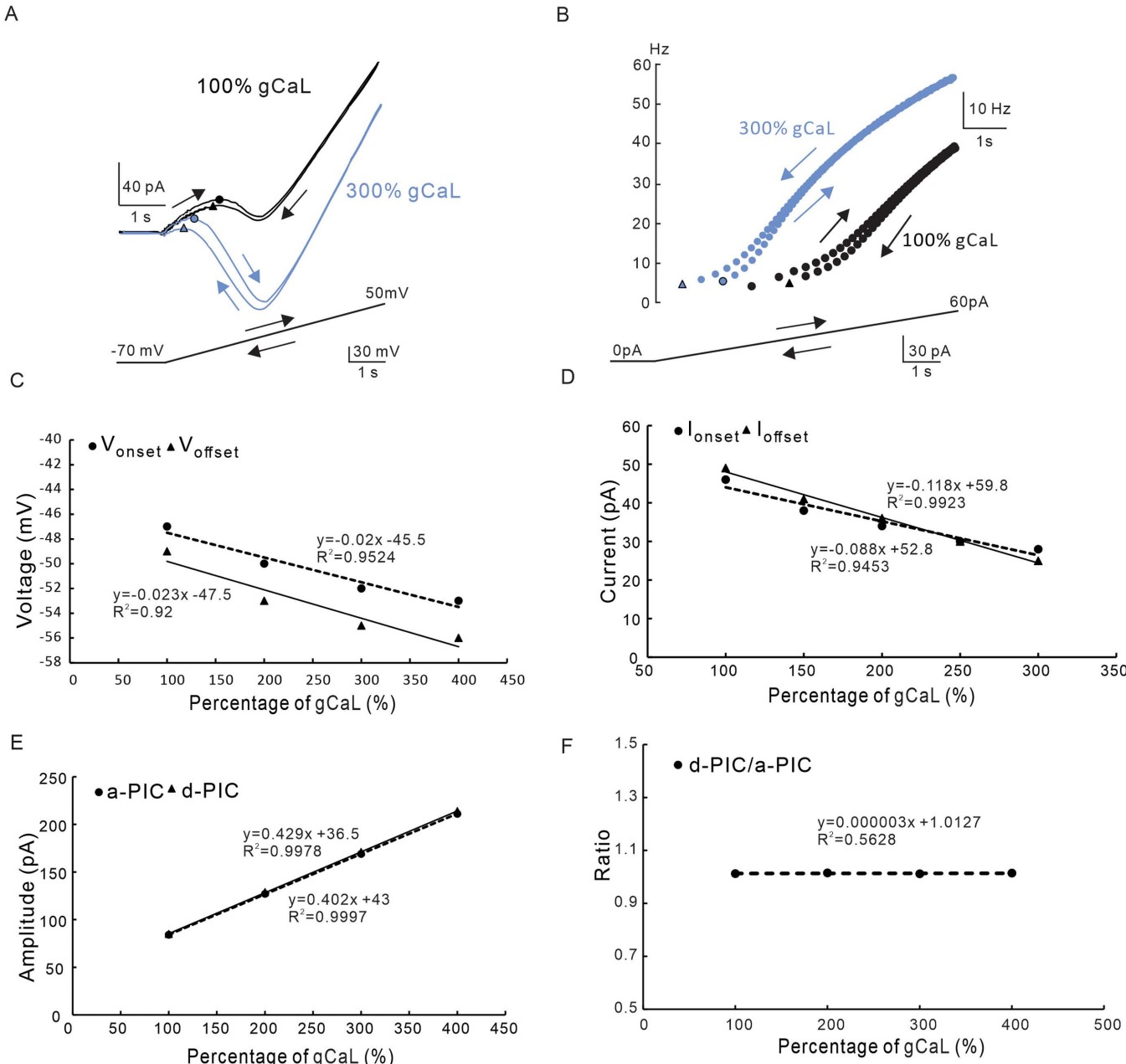

**Fig 6. The effect of CaL maximal conductance on PIC patterns and firing types. A.** PICs were induced by bi-ramps voltage (bottom) in 100% (black, control) and 300% (blue) of gCaL. **B.** Instantaneous firing frequency produced by 100% (black, control) and 300% (blue) of gCaL. **C.** The relationship between the $V_{onset}$ & $V_{offset}$ and percentage of gCaL. Increasing the gCaL hyperpolarized the $V_{onset}$ and $V_{offset}$ with almost unchanged $\Delta V$. **D.** The relationship between the $I_{onset}$ & $I_{offset}$ and percentage of gCaL. Increasing the gCaL reduced the $I_{onset}$ and $I_{offset}$ with a small change in $\Delta I$ from $\Delta I>0$ to $\Delta I<0$. **E.** The relationship between the amplitudes of a-PIC & d-PIC and percentage of gCaL. Increasing the gCaL increased amplitudes of a-PIC and d-PIC. **F.** The relationship between the ratio of d-PIC/a-PIC and gCaL. Increasing the gCaL did not significant change ratio of d-PIC/a-PIC amplitudes. The closed circle symbols stand for $V_{onset}$, $I_{onset}$ and a-PIC, and the closed triangular symbols represent $V_{offset}$, $I_{offset}$ and d-PIC.

type of the neuron. Simulation results showed that increasing gCaL conductance led to the prolongation of neuronal firing (Fig 6B). A further analysis indicated that increasing gCaL produced a linear hyperpolarization of $V_{onset}$ and $V_{offset}$ (Fig 6C) and changed the firing type of the neuron from $\Delta I > 0$ ($I_{onset} < I_{offset}$) to $\Delta I < 0$ ($I_{onset} > I_{offset}$). These results indicated that the neuronal firing could be prolonged after increasing the gCaL over 250% (Fig 6D). Increasing gCaL also linearly increased the amplitude of a-PIC and d-PIC (Fig 6E). However, the ratio of PIC amplitudes (d-PIC/a-PIC) remained almost unchanged (Fig 6F). These results suggested that the CaL mainly contributed to amplitude of PICs, delayed-inactivation of PICs and prolonged-firing of 5-HT neurons.

**Modulation of CaL kinetics.** Similar to the study of NaP, we also studied the effects of modulating kinetics of CaL on PIC patterns and firing properties with NaP conductance set to 0 (gNaP=0). Increasing the slope ($V_S$) of activation kinetics of CaL increased the $V_{onset}$ and $V_{offset}$ (Fig 7A and 7B) and reduced the $I_{onset}$ and $I_{offset}$ (Fig 7C). A further analysis showed that increasing $V_S$ of CaL produced a nonlinear depolarization of the $V_{onset}$ and $V_{offset}$ (Fig 7D) with $\Delta V$ changing from $\Delta V < 0$ to $\Delta V = 0$. Increasing the $V_S$ of CaL also reduced the $I_{onset}$ and $I_{offset}$ nonlinearly and changed the $\Delta I$ from $\Delta I > 0$ to $\Delta I = 0$, suggesting that increasing $V_S$ of CaL extended the band of repetitive firing (Fig 7C and 7E). As $V_S$ increased the amplitudes of a-PIC and d-PIC showed a nonlinear reduction (Fig 7F), while the ratio of PIC amplitudes increased linearly (Fig 7G). These results suggest that increasing the $V_S$ of CaL facilitated the clockwise trajectory of the PICs, increased the $\Delta V$ and reduced the $\Delta I$.

**The dendritic distribution of CaL.** The CaL channels mainly distributed in the dendrites of neurons [26,36,37]. Our recent studies reported that chronic exercise promoted dendritic growth in spinal and brainstem neurons and increased neuronal excitability [27,28,38]. These studies suggested that the dendrites of 5-HT neurons were extendable during chronic physiological activity. In the following simulation we tested the effect of dendritic distribution of CaL on the PIC patterns and firing types. When we increased the dendritic length, we also extended the CaL distribution to the new dendritic compartments. This increased total CaL conductance in dendrites since the density of CaL conductance in dendritic compartment remained unchanged (Table 3). The total CaL conductance in dendrites increased in proportion to increase of the dendritic length.

Simulation results demonstrated that increasing the length of dendrites induced the plateau potential in the dendrites (Fig 8A1), depolarized $V_{onset}$ and hyperpolarized $V_{offset}$ in soma (Fig 8A2). These results indicated that PICs activated (onset) at more depolarized membrane potential but terminated (offset) at more hyperpolarized membrane potential when dendritic length increased. Increasing dendritic length also increased $I_{onset}$ and lowered $I_{offset}$, suggesting that the repetitive firings delayed as well as extended (Fig 8B). We analyzed the changes in the $V_{onset}$, $V_{offset}$, $I_{onset}$ and $I_{offset}$ throughout the range of increasing the dendritic length (Fig 8C and 8D). Simulation results showed that increasing dendritic length depolarized the $V_{onset}$, hyperpolarized the $V_{offset}$ linearly, and reduced the $\Delta V$ dramatically (Fig 8C, note $\Delta V < 0$). Also, increasing dendritic length led to a rising of $I_{onset}$ and a nonlinear lowering of $I_{offset}$ and dramatically reduced $\Delta I$ (Fig 8D, note $\Delta I < 0$) which remarkably postponed the repetitive firings as shown in Fig 8B. Furthermore, extending the dendritic length increased the amplitudes of a-PIC and d-PIC linearly with almost the same amount of increment (Fig 8E) as well as unchanged the ratio of PIC amplitudes (Fig 8F). These results suggested that increasing dendritic length increased the membrane potential at which PICs activated (onset), lowered membrane potential at which PICs terminated (offset), and extended posterior firing (postponed firing). The extension of dendritic length did not change the direction of trajectory of PICs.

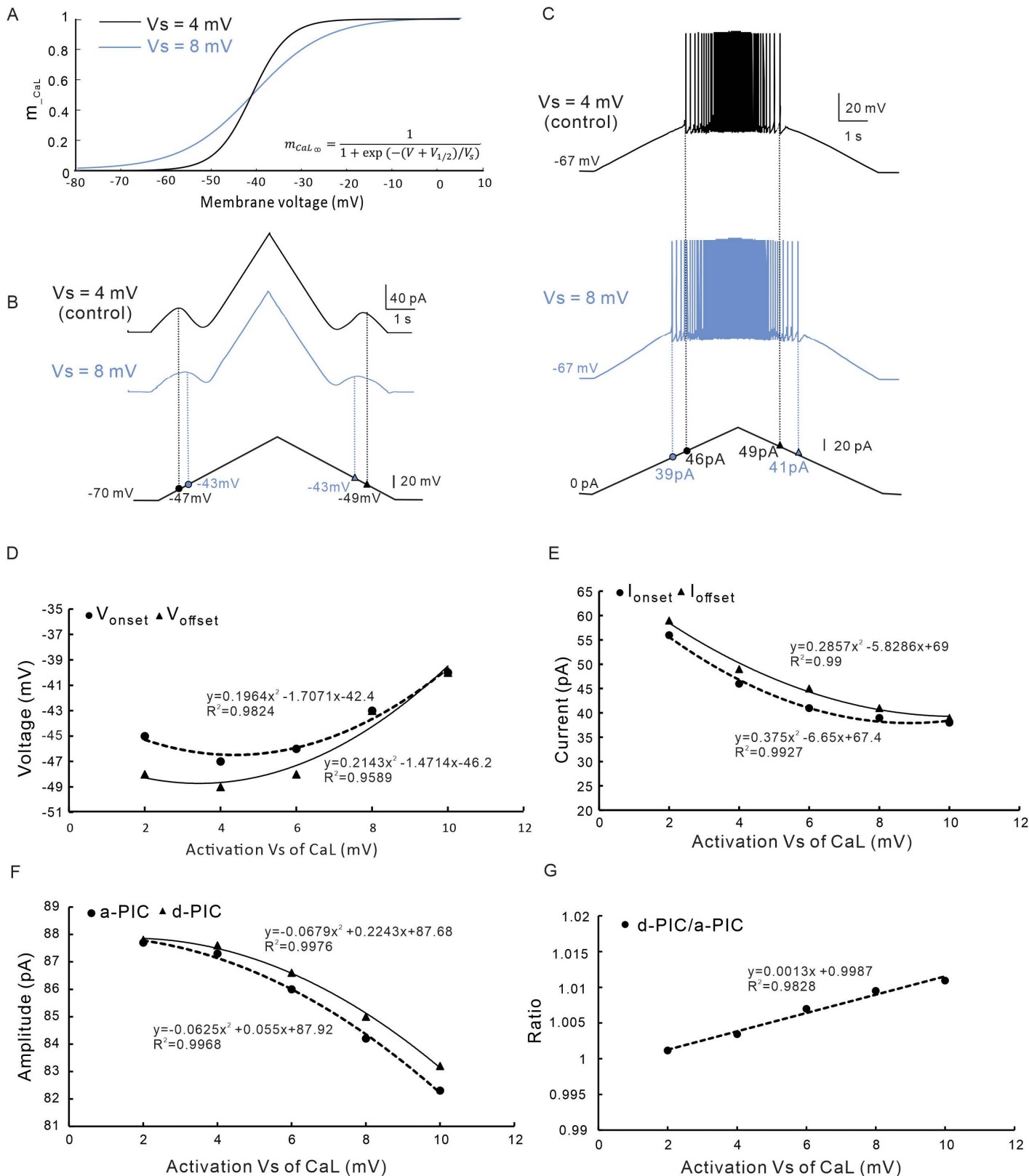

**Fig 7. The effect of activation variable $V_s$ of CaL on PIC patterns and firing types. A.** The activation gating ($m_{CaL}$) curve of CaL was established when $V_s$ was set to 4 (black, control) and 8 mV (blue), respectively. **B.** PICs were induced by bi-ramp voltage when $V_s$ of CaL was set to 4 (black, control) and 8 mV (blue),

respectively. **C.** Repetitive firings were evoked with bi-ramp current when $V_S$ was set to 4 (black, control) and 8 mV (blue), respectively. **D.** The relationship between the $V_{onset}$ & $V_{offset}$ and $V_S$ of CaL. Increasing the $V_S$ non-linearly depolarized the $V_{onset}$ and $V_{offset}$ with an increase in $\Delta V$. **E.** The relationship between the $I_{onset}$ & $I_{offset}$ and $V_S$ of CaL. Increasing the $V_S$ non-linearly reduced the $I_{onset}$ and $I_{offset}$ with a decrease in $\Delta I$. **F.** The relationship between the amplitude of a-PIC & d-PIC and $V_S$ of CaL. Increasing the $V_S$ non-linearly reduced amplitudes of a-PIC and d-PIC. **G.** The relationship between the ratio of d-PIC/a-PIC and $V_S$ of NaP. Increasing the $V_S$ increased ratio of d-PIC/a-PIC amplitudes. The closed circle symbols stand for $V_{onset}$, $I_{onset}$ and a-PIC, and the closed triangular symbols represent $V_{offset}$, $I_{offset}$ and d-PIC.

## Combined effects of modulating NaP and CaL on PIC patterns and firing types

In the following simulations we studied the combined effects of modulating $V_S$ and $S_{gate}$ of NaP on PIC patterns and firing types with CaL blocked (gCaL= 0). Fig 9 showed the effect of NaP kinetics on the $V_{onset}$, $V_{offset}$, and $\Delta V$ under the voltage-clamp and the $I_{onset}$, $I_{offset}$ and $\Delta I$ under the current-clamp modes, respectively. The $V_S$ dominated the $V_{onset}$ (Fig 9A1), while the $S_{gate}$ had little effect on the $V_{onset}$ (Fig 9A1). Similarly, $V_S$ mainly controlled the $I_{onset}$ while the $S_{gate}$ slightly affected the $I_{onset}$ (Fig 9B1). Both $V_{onset}$ and $I_{onset}$ decreased with increasing the value of $V_S$ of NaP, indicating that increasing $V_S$ lowered the membrane potential for PIC onset and reduced the current for repetitive discharge. We further analyzed the effects of regulating $V_S$ and $S_{gate}$ of NaP on $V_{offset}$ and $I_{offset}$. In general, Increasing $S_{gate}$ and $V_S$ hyperpolarized $V_{offset}$ and decreased $I_{offset}$ (Fig 9A2 and 9B2). The $V_{offset}$ hyperpolarization indicated a longer duration of the d-PIC, while the $I_{offset}$ reduction implicated a prolonged firing of the neuron. Simulation results showed that both $\Delta V$ and $\Delta I$ decreased when $S_{gate}$ increased and $V_S$ decreased (Fig 9A3 and 9B3). In this case, the duration of PICs was extended and the repetitive firing was prolonged, leading to an increased neuronal excitability.

The L-type calcium channels (CaL) are primarily distributed in the dendrites of neurons and play an essential role in amplifying synaptic currents and regulating neuronal excitability [34,39,40]. In the following simulation we investigated the mixed effects of modulating $V_{1/2}$, $V_S$ and dendritic distribution of CaL on the PIC patterns, firing properties and neuronal excitability. We focused on the changes in $V_{onset}$, $V_{offset}$, and $\Delta V$ under voltage-clamp mode and $I_{onset}$, $I_{offset}$ and $\Delta I$ under current-clamp mode, respectively. The NaP channels were blocked (gNaP = 0) in the following simulations. A hyperpolarization of $V_{1/2}$ represented a lower onset of CaL activation, and an increase of $V_S$ indicated a slower opening rate of CaL channels. A dendritic extension implicated a development of neuronal morphology which could be induced by exercise or cell development, while a dendritic reduction represented a decreased electronic length of dendrites which could be produced by channel regulation or neurotransmitter modulation.

In the following simulations we used a four-dimensional variable function to study the mixed effects of modulating $V_{1/2}$, $V_S$ and dendritic length on the PIC patterns (Fig 10A) and firing types (Fig 10B). Simulation results indicated that reducing dendritic length, increasing $V_S$ and hyperpolarizing $V_{1/2}$ induced a hyperpolarization of $V_{onset}$ (Fig 10A1) and $V_{offset}$ (Fig 10A2), which extended duration of PICs in 5-HT neurons thus increased neuronal excitability. Similarly, the same conditions reduced the $I_{onset}$ (Fig 10B1) but increased $I_{offset}$ (Fig 10B2), implicating that the neurons discharged earlier and terminated earlier, as well. A further analysis indicated that increasing dendritic length, reducing $V_S$ and depolarizing $V_{1/2}$ generally reduced the $\Delta V$ from 2 to $-8$ mV (Fig 10A3), suggesting that the patterns of PICs were changeable between $\Delta V > 0$ and $\Delta V < 0$, depending on the modulation of CaL kinetics and neuronal morphology. Similarly, increasing dendritic length, reducing $V_S$ and depolarizing $V_{1/2}$ reduced the $\Delta I$ from 10 to $-70$ pA, implicating that the types of repetitive firing were convertible between $\Delta I > 0$ and $\Delta I < 0$, depending on the CaL kinetics modulation and CaL channel distribution.

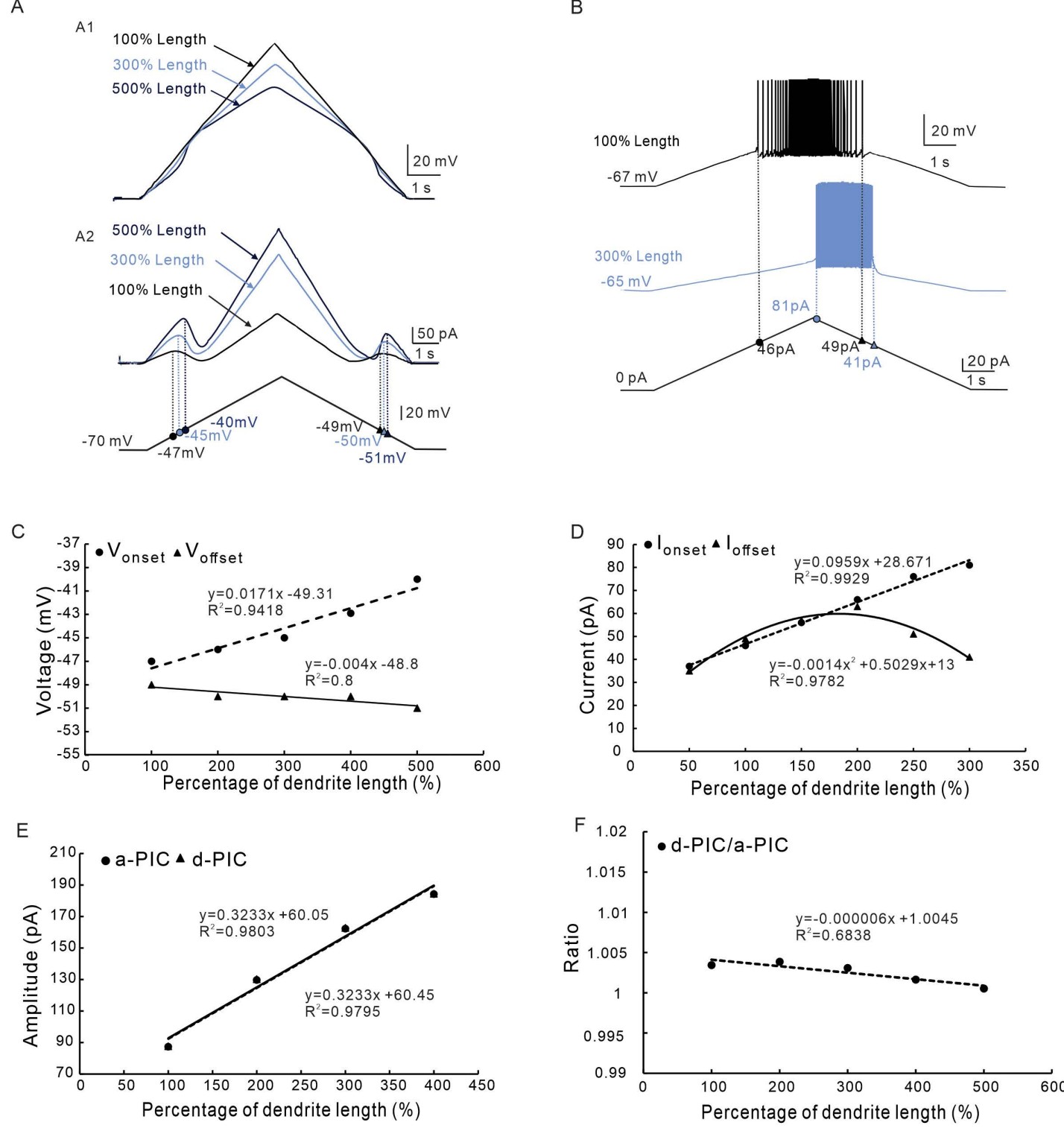

**Fig 8. The effect of dendritic distribution of CaL on PIC patterns and firing types. A.** PICs were induced by bi-ramp voltage when the dendritic length was extended to 100% (black, control), 300% (blue), and 500% (dark blue) with distal distribution of CaL. A1: Membrane potential measured at middle of dendrites when dendritic length was extended to 100% (control), 300%, and 500%. A2: PIC currents measured at the soma when dendrites length was extended to 100% (control), 300%, and 500%. **B.** Repetitive firings were evoked with current clamps when the dendritic length was extended to100% (control, black), and 300% (blue) with distal distribution of CaL. **C.** The relationship between the $V_{onset}$ & $V_{offset}$ and percentage of dendrite length. Increasing the dendrite length depolarized the $V_{onset}$ and hyper-polarized the $V_{offset}$ with a decrease in ΔV. **D.** The relationship between the $I_{onset}$ & $I_{offset}$ and percentage of dendrites length. Increasing the dendritic length increased the

$I_{onset}$ and reduced $I_{offset}$ non-linearly with a decrease in $\Delta I$. **E.** The relationship between the amplitude of a-PIC & d-PIC and percentage of dendrite length. Increasing the dendritic length simultaneously increased amplitudes of a-PIC and d-PIC. **F.** The relationship between the ratio of d-PIC/a-PIC and percentage of dendrite length. Increasing the dendritic length slightly reduced ratio of d-PIC/a-PIC amplitudes. The closed circle symbols stand for $V_{onset}$, $I_{onset}$ and a-PIC, and the closed triangular symbols represent $V_{offset}$, $I_{offset}$ and d-PIC.

Previous studies suggested that a lowering of $V_{onset}$ and $I_{onset}$ could make the 5-HT neurons more easily activated by synaptic currents from the midbrain, thus enhancing their functional role in generating locomotor activity [18,23,41]. Furthermore, a reduction of $V_{offset}$, $I_{offset}$, $\Delta V$, and $\Delta I$ could prolong the repetitive firing of 5-HT neurons and thus increasing the neuronal excitability. In summary, modulation of CaL channels are essential for regulating excitability and facilitating function of 5-HT neurons in serotonergic system [7,20,24].

## Discussion

In this study, we reported six patterns of PICs in medullary 5-HT neurons. These patterns can be divided into clockwise (a-PIC < d-PIC) and counterclockwise (a-PIC > d-PIC) trajectory according to the amplitudes of a-PIC and d-PIC, respectively, and three categories corresponding to the values of $\Delta V$, i.e., $\Delta V > 0$, $\Delta V = 0$ and $\Delta V < 0$, respectively. We also demonstrated three types of repetitive firing of 5-HT neurons based on the values of $\Delta I$, i.e., $\Delta I > 0$, $\Delta I = 0$ and $\Delta I < 0$. We studied the contributions of NaP, CaL and neuronal morphology to the PIC patterns and discharge types. Our results indicated that NaP regulated PICs with counterclockwise trajectory and $\Delta V > 0$ while CaL determined PICs with clockwise trajectory and $\Delta V < 0$ and contributed to the prolonged firing of the neurons ($\Delta I < 0$). The novel mechanism we discovered in this study was that the distribution of CaL in distal dendrites controlled the patterns of PICs and types of repetitive firings in 5-HT neurons.

### Impact of distal distribution of CaL on prolonged firing

CaL are mainly distributed on the dendrites of spinal neurons in rodents [28,42,43]. Similar results are also observed in 5-HT neurons of mouse brainstem [27]. In this study we specifically investigated the regulatory effect of increasing the dendrites on PIC patterns and firing types. Increasing the dendritic length, which led to more distal distribution of CaL, produced a significant depolarization of the $V_{onset}$, hyperpolarization of the $V_{offset}$, and increase of the $\Delta V$ (Fig 8A and 8C). This implicated that the inactivation of PICs was significantly enhanced. However, the clockwise trajectory of PICs did not change substantially with increasing the dendrites (Fig 8F). A novel finding in this study was that the types of repetitive firing changed dramatically with the increase of dendritic length (Fig 8B), which was manifested by a significant increase in $I_{onset}$ and substantial decrease in $I_{offset}$ (Fig 8D). A dramatically postponed firing was induced in this case (Fig 8B). The fact was that the distal distribution of CaL in dendrites postponed firing of 5-HT neurons.

In studies of spinal motoneurons, the difference of instantaneous frequency ($\Delta F$) between recruitment and de-recruitment of currents was commonly used to represent the delay of discharge [44–46]. It was found that PICs was a primary factor determining the neuronal $\Delta F$, particularly when CaL channels were distributed on neuronal dendrites [34,44,47,48]. In this case, the de-recruitment could occur at a lower input level than the recruitment ($\Delta I < 0$) in spinal motoneurons [34]. Our data was consistent with these results and confirmed the contribution of dendritic CaL to the prolonged firing.

### CaL contribution to PIC patterns and firing types

It was found in previous studies that CaL dominated the major component of the PICs [1,3,21,49] and primarily distributed in the dendrites of spinal neurons with a higher

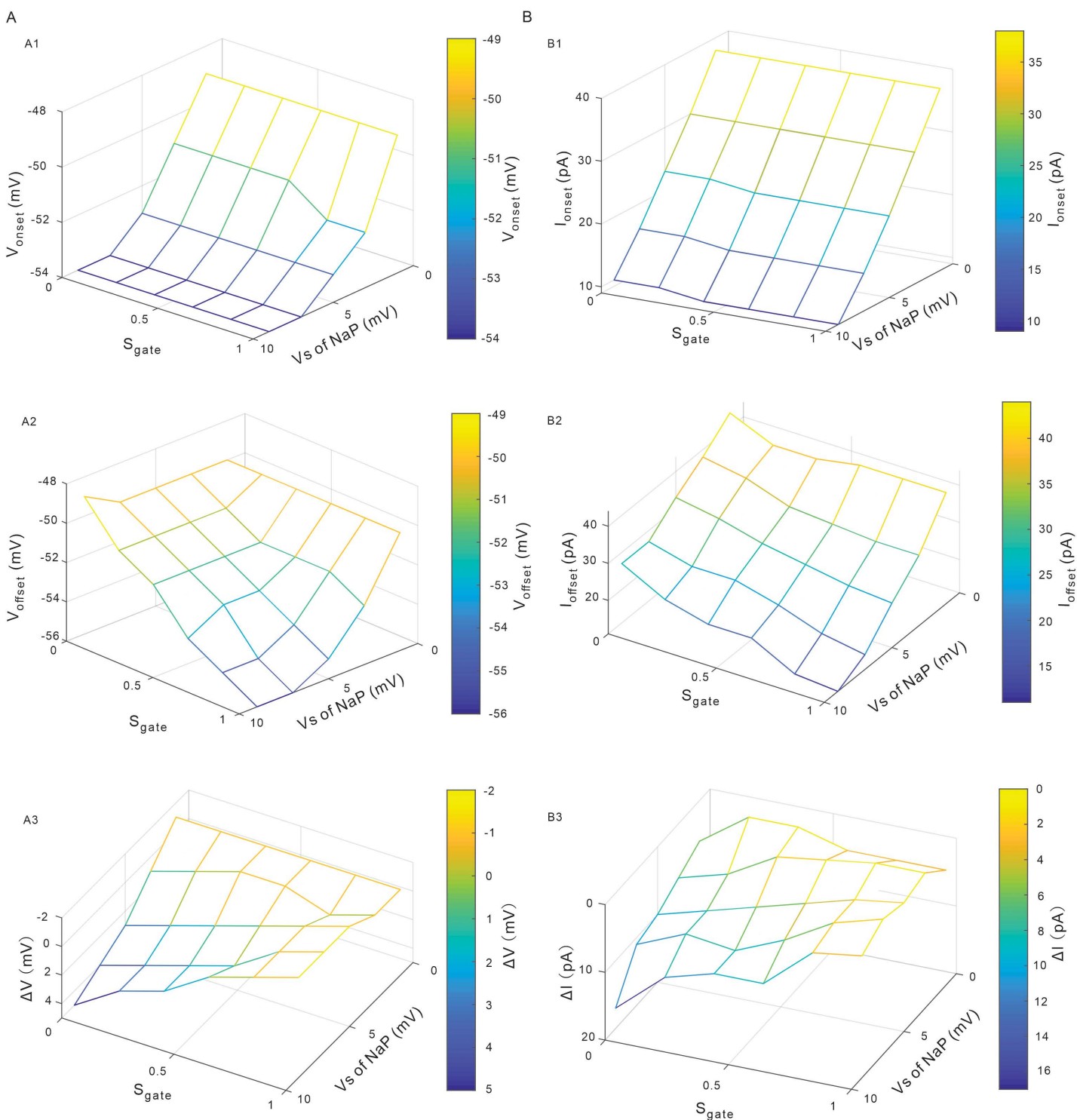

**Fig 9. Combined effects of modulating $V_S$ and $S_{gate}$ of NaP on PIC patterns and firing types. A.** Combined effects of modulating $V_S$ and $S_{gate}$ of NaP on $V_{onset}$ (A1), $V_{offset}$ (A2) and $\Delta V$ (A3), respectively. **B.** Combined effects of modulating $V_S$ and $S_{gate}$ of NaP on $I_{onset}$ (B1), $I_{offset}$ (B2) and $\Delta I$ (B3), respectively.

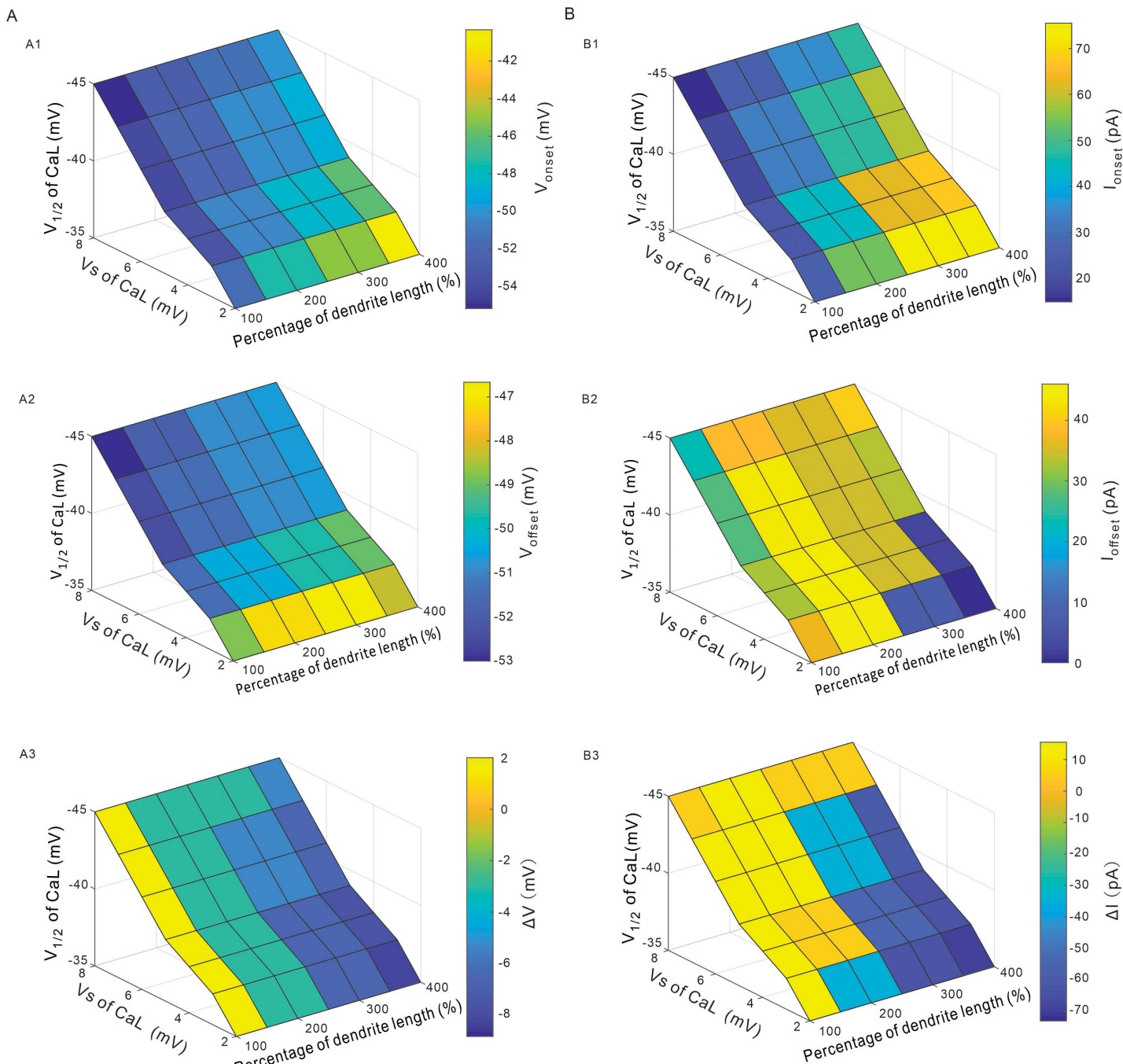

**Fig 10. Combined effects of $V_{1/2}$, $V_S$ of CaL and dendritic length on PIC patterns and firing types. A.** Combined effects of modulating $V_{1/2}$, $V_S$ and dendritic length on $V_{onset}$ (A1), $V_{offset}$ (A2) and $\Delta V$ (A3), respectively. **B.** Combined effects of modulating $V_{1/2}$, $V_S$ and dendritic length on $I_{onset}$ (B1), $I_{offset}$ (B2) and $\Delta I$ (B3), respectively.

expression level than NaP [50–52]. In previous modeling study we demonstrated that increasing the maximum conductance of gCaL by 150% induced changes in the frequency-current (f-I) relationship of motoneurons that closely resembled the changes observed in spinal motoneurons during locomotion [53]. On the other hand, however, the same amount alteration of gNaP produced less effect than CaL on the f-I relationship, suggesting a dominant role and

wider modulation range for CaL channels. The above studies provided us with a physiological basis for modulation of CaL and NaP in the present study.

The CaL channels play multiple functional roles in spinal neurons including regulation of neuronal discharge, generation of plateau potential, amplification of synaptic currents, maintenance of bistable firing, and plasticity of locomotor system [33,37,54–56]. CaL is also shown to generate the clockwise trajectory of PICs in spinal neurons [37]. In recent study we demonstrated that blocking CaL stopped the prolonged firing in medullary 5-HT neurons of mice, suggesting that CaL dominated the prolonged firing [20]. These studies have unveiled the essential role of CaL in generating multiple patterns of PICs and different types of firings in rodents. In this study, we focused the maximum conductance (gCaL) and kinetics ($V_S$ and $V_{1/2}$) of CaL on formation of PIC patterns and firing types in 5-HT neurons. Modulation of gCaL altered the $V_{onset}$ and $V_{offset}$ parallelly without substantially changing $\Delta V$ and clockwise trajectory of PICs (Fig 6A and 6C). Similarly, modulation of gCaL changed the $I_{onset}$ and $I_{offset}$ simultaneously with small change in $\Delta I$ (Fig 6B and 6D). These results suggested that modulation of gCaL mainly contributed to amplitude of PICs (Fig 6E) and frequency of firing (Fig 6B) rather than the patterns of PICs and types of firing in 5-HT neurons.

On the contrary, however, modulation of CaL kinetics altered the patterns of PICs and firing properties. Reducing $V_S$ decreased the values of $\Delta V$ and $\Delta I$ from positive to negative with dendritic length increased from 100% to 400% (Figs 7D, 7E, 10A3 and 10B3). Similarly, depolarizing $V_{1/2}$ reduced the values of $\Delta V$ and $\Delta I$ from positive to negative with increasing the dendritic length (Fig 10A3 and 10B3). In this case, the types of repetitive firing changed from type 1 ($\Delta I > 0$) to type 2 ($\Delta I = 0$) and finally to type 3 ($\Delta I < 0$), i.e., the prolonged repetitive firing (Fig 2B). These results suggested that modulation of CaL kinetics ($V_s$ and $V_{1/2}$) determined the delayed-inactivation of PICs (Fig 7A and 7B) and changed the firing types of 5-HT neurons (Fig 7C and 7E). These results are consistent with previous studies on calcium currents in spinal neurons [1,35].

## NaP contribution to PIC patterns and firing types

NaP is an essential component of PICs with smaller proportion than CaL in spinal neurons and brainstem 5-HT neurons in rodents [2,21,38]. NaP plays an important role in regulating neuronal threshold of discharge as well as rhythm of locomotion [9,25,27,57]. NaP also contributes to plasticity of locomotor system in physical exercise [58]. In this study, we investigated the effects of modulating the maximum conductance (gNaP) and kinetics ($V_S$ and $S_{gate}$) of NaP on the PIC patterns and firing types. Modulation of gNaP hyperpolarized the $V_{onset}$ and $V_{offset}$ parallelly without significant change in $\Delta V$ and counterclockwise trajectory of PICs (Fig 3A and 3C). Similarly, modulation of gNaP reduced the $I_{onset}$ and $I_{offset}$ simultaneously with almost unchanged $\Delta I$ (Fig 3B and 3D). These results implicated that modulation of gNaP mainly contributed to amplitude of PICs (Fig 3E) rather than the PIC patterns and firing types.

Different from gNaP, modulation of NaP kinetics affected the patterns of PICs and types of firing. Increasing $S_{gate}$ reduced $\Delta V$ from positive to negative (Figs 5A, 5C and 9A3) and $\Delta I$ from 16 to 0 (Figs 5B, 5D and 9B3). In this case, the trajectory of PICs changed from counterclockwise (a-PIC > d-PIC) to clockwise (a-PIC < d-PIC) and firing types from type 1 ($\Delta I > 0$) to type 2 ($\Delta I = 0$). These results suggested that $S_{gate}$ played an essential role in regulating the patterns of PICs and types of firing. On the other hand, however, modulation of $V_S$ of NaP did not substantially impact on the PIC patterns and firing types. Increasing $V_S$ increased $\Delta V$ (Figs 4B, 4D and 9A3) and $\Delta I$ (Figs 4B, 4E and 9B3) but did not significantly change the direction of PICs trajectory and firing types of the neurons. In conclusion, NaP mainly contributed to counterclockwise trajectory of PICs and types of firing through modulating slow

inactivation gate $S_{gate}$, whereas the maximum conductance gNaP and kinetics $V_S$ had a limited effect on PIC patterns and firing types in 5-HT neurons.

## Role of PICs in modulating neuronal excitability and locomotion

PICs play many functional roles in central nervous system, including amplifying synaptic currents, inducing plateau potentials and maintaining bistable firing in spinal neurons [6,34]. PICs also play a role in generating locomotion through facilitating recruitment of motoneurons, rhythmic generation and the strength of muscle contraction [12–14]. We explored the mechanism underlying different patterns of PICs and types of firing which were essential for modulating neuronal excitability and locomotion. These findings suggested that that NaP predominantly controlled the thresholds (onset & offset) of PICs in medullary 5-HT neurons, while CaL primarily dominated the amplitudes (a-PIC & d-PIC) of PICs. Both PICs played essential in roles in regulating neuronal excitability with different target values.

## Studies of PIC patterns and firing types in different neurons

The relationship between PIC patterns and neuronal firing properties have been studied intensively in spinal motoneurons [47,59,60]. The present study is the first to explore this relationship in brainstem 5-HT neurons which are essential for initiation of locomotion in rodents. Recent modeling study reported that distribution of CaL channels in the distal dendrite hyperpolarized $V_{offset}$ and enhanced the delay of PICs in spinal motoneurons [61]. Our simulation results supported this report. However, our data further showed that extension of dendrites drove the $V_{onset}$ to significantly depolarize. In this case, a delayed inactivation of PICs ($\Delta V < 0$) was induced (Fig 8A and 8C) with dramatically postponed firings ($\Delta I < 0$) (Fig 8B and 8D). Since 5-HT neurons have relatively shorter dendrites than spinal motoneurons [27], it might be the reason that motoneurons are more likely to produce PICs with a delayed-inactivation [37], while 5-HT neurons rarely express such pattern of PICs (Fig 2C) [21,24].

The PICs also play a major role in regulating neuronal activity, particularly in terms of maximum firing frequency, duration of neuronal firing and spiking initiation time and so on [5,10,25,37,62,63]. Previous studies have indicated that both NaP and CaL enhance neuronal firing frequency [1,57,64]. Specifically, CaL facilitates the generation of plateau potentials, thus prolonging the duration of neuronal firing [3,34,60,65], while NaP modulates the voltage and current thresholds for action potential generation thus regulates the spiking initiation time [20–21]. In this study, we unveiled the relationship between PIC patterns and firing types, specifically the effects of PICs on the recruitment ($I_{onset}$) and de-recruitment ($I_{offset}$) of currents.

## Simulation issues related to the model

The single-cell model was constructed based on the membrane properties of medullary 5-HT neurons recorded in our experiments (Table 1). In general, altering the maximum conductance or gating kinetics of NaP and CaL generated the PIC patterns and neuronal firing types that closely matched the experimental data [20,21,24]. However, there were some issues about the parameters we used for simulation in the model.

**Range of conduction and kinetics for modulation.**   The modulating ranges of conductance and kinetics in the NaP and CaL models were based on previous modeling studies as well as experimental data. Usually, the alteration ranges of conductances are set to 100–200% of control values (control defined as 100% in the present model) and the voltage ranges of kinetics to 2–6 mV in previous modeling studies [29,66]. In this study, however, we increased conductances up to 500% and kinetics of activation to 10 mV in some cases in

order to predict the trends of modulatory effects of NaP and CaL on PIC patterns and firing types. In fact, previous studies have reported that moderate intensity chronic exercise induced ~70% increase in the amplitudes of NaP and CaL (i.e., conductances) and ~5.0 mV decrease in the voltage onset of NaP and CaL (i.e., activation kinetics), respectively, in spinal neurons [28], suggesting that NaP and CaL conductances and their kinetics were changeable during exercise and that the degree of changes could depend on the intensity and performance of the exercise. Previous studies further showed that chronic exercise promoted dendritic growth by ~70% in spinal and brainstem neurons and led to an increased neuronal excitability[27,28,38]. These results provided a physiological basis for modulating of NaP/CaL and dendritic length. In the present study, we altered PIC conductances and dendritic length in proportion to the experimental data. As to the changes in dendritic length, the prolonged firing occurred ($\Delta I <$ 0) when the dendritic length increased over 150% (Fig 8D). This effect became more dramatic with further increasing the dendrites (Fig 8B).

**Parameters of kinetics for modulation of NaP.** $V_s$ determines the sensitivity of NaP channel to membrane potential while the time constant $\tau_s$ regulates the time for the channel opening. Since NaP is a voltage-gated persistent opening channel, modulating $V_s$ produces bigger effect on PIC patterns and firing types than modulating $\tau_s$. For this reason, we selected $V_s$ rather than $\tau_s$ as a major parameter for modulation of NaP kinetics in this study.

## Conclusion

PICs exhibited multiple patterns with various firing types in medullary 5-HT neurons of mice. NaP conductance contributed to amplitude of PICs, whereas the gating property of slow inactivation of NaP regulated the PIC patterns and firing types. CaL conductance dominated the amplitude of PICs, while CaL kinetics determined inactivation of PICs and prolongation of repetitive firing. Distribution of CaL in distal dendrites determined the patterns of PICs and types of repetitive firings in 5-HT neurons.

## Materials and methods

### Ethics statement

Experiments were carried out in accordance with the East China Normal University Laboratory Animal Center, and all procedures were in accordance with protocols approved by the Animal Experiment Ethics Committee (Ethics No. ARXM2024053).

### Animal model

The experiments were carried out on neonatal *ePet-EYFP mice* (3–6 days old), generated by crossing *ePet-cre mice* (The Jackson Laboratory, stock no. 012712) with *R26-stop-EYFP mice* (The Jackson Laboratory, stock no. 006148). Animals were exposed to a 12:12-h light-dark cycle and had free access to food and water. Their pain and distress were minimized.

### Preparation of slices and patch-clamp recordings

The general experimental and surgical procedures have been described in detail in previous studies [20]. The 3- to 6-day-old (P3–P6) *ePet-EYFP mice* of either sex were euthanized by cervical dislocation and quickly decapitated. To study 5-HT neurons, a section of medulla was removed and glued to a Plexiglas tray filled with cooled dissecting artificial cerebrospinal fluid (ACSF) bubbled with 95% $O_2$+5% $CO_2$. 250 μm-thick transverse slices of medulla were cut throughout the length of the 5-HT nucleus, transferred to a holding chamber, and incubated at room temperature (20–22°C) for 30-min recovery in recording ACSF. Slices were

transferred to a recording chamber mounted in the stage of an upright Olympus BX50 microscope fitted with differential interference contrast optics and epifluorescence. The chamber was perfused with recording ACSF at rate of 2 mL/min, bubbled with 95% $O_2$+5% $CO_2$. The enhanced yellow fluorescent protein (EYFP) neurons located in the medulla were identified at X 40 magnification by epifluorescence with a narrow-band yellow fluorescent protein cube (Fig 1A). The visualized EYFP neurons were patched with glass pipette electrodes. The pipette electrodes were pulled from borosilicate glass (1B150F-4; WPI) with an electrode puller (P-1000; Sutter Instrument) and had resistances of 6–8 MΩ when filled with intracellular solution (see Solutions). A MultiClamp 700B, a Digidata 1550, a MiniDigi 1B, and pCLAMP (10.7) (all from Molecular Devices) were used for data acquisition. Data were low-pass filtered at 3 kHz and sampled at 10 kHz. Whole cell patch recordings were made in voltage-clamp mode with capacitance compensation and in current-clamp mode with bridge balance. Electrophysiological data were analyzed with Axon Clampfit (10.7).

## Measurement of PIC parameters

PICs were recorded by applying a family of 5 slow bi-ramps voltage (10-s duration, 40 mV step of peak) to the neurons. Normally, the best of the recording was chosen to calculate the PIC parameters. After leak subtraction, a straight line (dashed line in Fig 1B) was drawn alone the rising phase of the current trace. The last point where the straight line touched the rising current trace was defined as the onset of PIC, and the corresponding voltage on the bi-ramp voltage at this point was defined as the onset voltage of PIC ($V_{onset}$). The lowest point on the ascending current trough was defined as the amplitude of the a-PIC. The first point where the straight line was tangent to the descending current trace was defined as the offset of PIC and the corresponding voltage on the bi-ramp voltage was defined as the offset voltage of PIC ($V_{offset}$). The lowest point on the descending current trough was defined as the amplitude of the d-PIC. And then we calculated the difference $\Delta V = V_{offset} - V_{onset}$. Details of the measurement are shown in Fig 1B.

In order to investigate the contribution of PICs to the regulation of neuronal firing properties, we recorded 5-HT neurons in current clamp mode, where a slow bi-ramp current with a duration of 10 s, peak of 60–80 pA, and holding current of 0 pA was applied to the neurons. The instantaneous frequency of firing was calculated. The onset current ($I_{onset}$) was defined as the point of the depolarizing current ramp at which the first spike was initiated, and the offset current ($I_{offset}$) as the point of the repolarizing current ramp at which the last spike was generated. And then we calculated the difference $\Delta I = I_{offset} - I_{onset}$ (Fig 1C). Cells selected for data analysis must meet the following conditions: stable resting membrane potential between −55 and −70 mV, input resistance ≥ 300 MΩ, action potential amplitude ≥ 40 mV, and time for intracellular recording ≥ 20 min.

## Solutions

**Dissecting ACSF.** Dissecting ACSF contained (in mM) 25 NaCl, 253 sucrose, 1.9 KCl, 1.2 $NaH_2PO_4$, 10 $MgSO_4$, 26 $NaHCO_3$, 1.5 kynurenic acid, 25 glucose, and 1.0 $CaCl_2$.

**Recording ACSF for voltage clamp.** Recording ACSF for voltage clamp contained (in mM) 125 NaCl, 2.5 KCl, 26 $NaHCO_3$, 1.25 $NaH_2PO_4$, 25 glucose, 1 $MgCl_2$, 10 tetraethylammonium chloride (TEA-Cl), and 2.0 $CaCl_2$.

**Recording ACSF for current clamp.** Recording ACSF for current clamp contained (in mM) 125 NaCl, 2.5 KCl, 26 $NaHCO_3$, 1.25 $NaH_2PO_4$, 25 glucose, 1 $MgCl_2$, and 2.0 $CaCl_2$.

**Intracellular solution for voltage clamp.** The intracellular solution for voltage clamp contained (in mM) 135 K-gluconate, 10 NaCl, 20 TEA-Cl, 10 HEPES, 2 $MgCl_2$, 5 Mg-ATP and 0.5 GTP.

**Intracellular solution for current clamp.** The intracellular solution for current clamp contained (in mM) 135 K-gluconate, 10 NaCl, 10 HEPES, 2 MgCl$_2$, 5 Mg-ATP and 0.5 GTP.

The pH of these solutions was adjusted to 7.3 with HCl. Osmolarity was adjusted to 305 mosM by addition of sucrose to the solution.

## 5-HT neuron model

A 5-HT neuron model was built with NEURON, based on the membrane properties of 5-HT neurons of brainstem (Fig 1D) and modified from motoneuron model [67]. The structure of the 5-HT neuron model had dendrite, soma, initial segment (IS), and axon cable. The dendrite consisted of 25 compartments whose diameters first increased linearly from 1.4 μm to 5.4 μm with distance from the soma over the 20% of dendrite length and then decreased linearly from 5.4 μm to 0 μm over the remainder of dendrite length. The soma was a single compartment and the IS 7 compartments whose diameters decreased linearly from 8.2 μm to 1.8 μm from soma to axon. The axon consisted of 9 compartments with equal diameter (see Fig 1D and Table 2). The soma surface area was 753 μm², and the dendrite length was set as 600 μm. The model includes six active conductances including transient fast sodium (NaT); persistent sodium (NaP); delayed-rectifier potassium (Kdr); calcium-activated potassium (KCa) L-type calcium conductance (CaL) and leak conductance. Details of the distribution were shown in Table 2.

The model cell was defined as a mouse 5-HT neuron in the medulla. The membrane properties were comparable to the physiological experiment data, and the target properties consisted of the resting membrane potential (E$_m$), current threshold (I$_{th}$), voltage threshold (V$_{th}$), action potential (AP) height and 1/2 width, afterhyperpolarization (AHP) amplitude and 1/2 width, and input resistance (R$_{in}$). Detail of the properties of the neuron and model were summarized in Table 1.

The cable equation for all compartments in model can be written as:

$$C_m \frac{dV_m}{dt} = -I_L - I_{ion} - I_{coupling} - I_{syn} + I_{inj}$$

Where $C_m$ was membrane capacitance; $I_L$ was potassium mediated leak current; $I_{ion}$ was ionic current of active conductance; $I_{coupling}$ was current from adjacent compartments; $I_{syn}$ was synaptic current; $I_{inj}$ was injected current. The ionic currents consisted of NaT, Kdr, NaP, CaL, and KCa. All channels mediating these currents were defined by a Hodgkin-Huxley equation:

$$\frac{dP}{dt} = \alpha(1-P) - \beta P$$

where steady-state value $P_\infty = \alpha / (\alpha + \beta)$ and time constant $\tau = 1/(\alpha + \beta)$.

The transient Na ionic current ($I_{NaT}$) was described by the following equations:

$$I_{NaT} = g_{NaT} \cdot m_{NaT}^{3} \cdot h_{NaT} \cdot (V - V_{Na})$$

$$\alpha_m = \frac{0.4 \cdot (V + 20)}{1 - \exp[-(V + 20)/7.2]}$$

$$\beta_m = \frac{0.124 \cdot (V + 20)}{1 - \exp[-(V + 20)/7.2]}$$

$$\tau_m = \frac{1}{(\alpha_m + \beta_m) * Q(T)}$$

$$m_{NaT\infty} = \frac{\alpha_m}{\alpha_m + \beta_m}$$

$$\alpha_h = \frac{0.06 \cdot (V + 25)}{1 - \exp[-(V + 25)/1.3]}$$

$$\beta_h = \frac{0.01 \cdot (V + 25)}{1 - \exp[-(V + 25)/1.3]}$$

$$\tau_h = \frac{1}{(\alpha_h + \beta_h) * Q(T)}$$

$$h_{NaT\infty} = \frac{1}{1 + \exp[(V + 40.5)/4.8]}$$

where $g_{NaT}$ was maximum conductance; $m_{NaT}$ and $h_{NaT}$ were the activation and inactivation gating variables; the equilibrium potential of transient Na ($V_{Na}$) was set to +50 mV; $\alpha_m$, $\beta_m$, $\alpha_h$, and $\beta_h$ were rate constants for activation and inactivation; $\tau_m$ and $\tau_h$ were time constant for activation and inactivation; temperature factor $Q(T)$ was set to 2.46, which based on 37°C.

The component of NaP was persistent Na current ($I_{NaP}$), which satisfied the following equations:

$$I_{NaP} = g_{NaP} \cdot m_{NaP} \cdot s_{NaP} \cdot (V - V_{Na})$$

$$m_{NaP\infty} = \frac{1}{1 + \exp(-(V + 37.3mV)/V_s)}$$

$$\tau_m = 1 \, ms$$

$$s_{NaP\infty} = s_{gate} + (1 - s_{gate}) \cdot \frac{\alpha_s}{(\alpha_s + \beta_s)}$$

$$\alpha_s = 0.001 \cdot \exp[-(V + 85)/30]$$

$$\beta_s = 0.034 \cdot \exp[-(V + 17)/10]$$

$$\tau_s = \frac{1}{\alpha_s + \beta_s}$$

where maximum conductances were $g_{Nap}$ for persistent Na current ($I_{NaP}$); equilibrium potential for persistent Na channels ($V_{Na}$) was set to +50 mV; $m$ and $s$ were activation and slow inactivation gating variables; the steady-state activation gating variables ($m_{NaP}$) were

defined by a sigmoidal curve; the slope of activation ($V_s$) was set to 5mV; slow inactivation was simulated using a model similar to that used by Fleidervish and colleagues [68]; $s_{gate}$ was the minimum value of the slow inactivation gate variable.

The delayed rectifier K current also simulated using a sigmoidal steady-state activation curve ($n_\infty$) and a voltage-dependent time constant ($\tau_n$), which was governed by the following equations:

$$I_{Kdr} = g_{Kdr} \cdot n_{Kdr}{}^4 \cdot (V - V_K)$$

$$\tau_n = 0.8 + 20 \frac{\exp\left[\dfrac{V + 39}{5.5}\right]}{\left\{1 + \exp\left[\dfrac{V + 39}{5.5}\right]\right\}^2}$$

$$n_{Kdr\infty} = \frac{1}{1 + \exp\left(-(V + 25)/20\right)}$$

where equilibrium potential for delayed rectifier K channels was set to −77 mV; n was an activation variable of the delayed rectifier.

The KCa was simulated using a model from Powers and colleagues [67] and initially described by Sah [69]. The Ca dynamics were represented by a first-order process:

$$\frac{dCa}{dt} = \frac{[Ca]_\infty - [Ca]}{\tau_{Ca}} - \frac{i_{Ca}}{2 \cdot F \cdot d}$$

where the time constant of decay ($\tau_{Ca}$) was set to 120 ms for KCa located in the distal dendrite, whereas the values of $\tau_{Ca}$ were given small value (76 ms) for the KCa channels governing the AHP; $F$ was Faraday's constant; $i_{Ca}$ was Ca current density; $d$ was the depth of KCa channels, which set 100 µm for the distal KCa and 0.6 µm for the KCa mediating the AHP. All KCa channels were described by the following equations:

$$I_{KCa} = g_{KCa} \cdot n_{KCa} \cdot (V - V_K)$$

$$\alpha_n = 0.1 \cdot \left([Ca] - [Ca_\infty]\right)^2$$

$$\beta_n = 0.1$$

$$\tau_n = \frac{1}{\alpha_n + \beta_n}$$

$$n_{KCa\infty} = \frac{\alpha_n}{\alpha_n + \beta_n}$$

The component of CaL providing the Ca current for KCa channels. The channel function satisfies the following equations:

$$I_{CaL} = g_{CaL} \cdot m_{CaL} \cdot (V - V_{CaL})$$

$$m_{CaL\infty} = \frac{1}{1 + \exp\left(-\left(V + V_{1/2}\right) / V_s\right)}$$

where the $g_{CaL}$ was the maximum conductance of CaL; $m_{CaL}$ was activation gating variables of CaL; The half activation voltage ($V_{1/2}$) was set to −41 mV; $V_s$ was the slope of activation, which was set to 5 ms in this study.

## Author contributions

**Conceptualization:** Yue Dai, Yi Cheng, Xingyu Wang.

**Funding acquisition:** Yue Dai.

**Investigation:** Yi Cheng, Renkai Ge.

**Methodology:** Xingyu Wang, Qiang Zhang.

**Supervision:** Yue Dai, Mei Zhou.

**Visualization:** Mei Zhou.

**Writing – original draft:** Yi Cheng, Xingyu Wang, Yue Dai.

**Writing – review & editing:** Yi Cheng, Yue Dai.

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
