## [Decision Letter · Decision Letter 0]

5 Sep 2024

Dear Professor Dai,

Thank you very much for submitting your manuscript "Multiple patterns of persistent inward currents with manifold types of repetitive firing in medullar serotonergic neurons of mice: the experimental and modeling study" for consideration at PLOS Computational Biology.

As with all papers reviewed by the journal, your manuscript was reviewed by members of the editorial board and by independent reviewers. In light of the reviews (below this email), we would like to invite the resubmission of a significantly-revised version that takes into account the reviewers' comments.

We cannot make any decision about publication until we have seen the revised manuscript and your response to the reviewers' comments. Your revised manuscript is also likely to be sent to reviewers for further evaluation.

Please revise the text and figures substantially according to the reviewers' comments, with particular emphasis on explaining the logic and rationale of your simulations. Please provide the code and missing information about your model (e.g. how it was tuned to the data) as requested by the reviewers.

Sincerely,

Peter Jedlicka

Guest Editor

PLOS Computational Biology

Lyle Graham

Section Editor

PLOS Computational Biology

Reviewer's Responses to Questions

**Comments to the Authors:**

Reviewer #1: Summary:

This work looks at the mechanistic impacts of different ion channel and morphology parameters on output electrophysiology, in particular persistent inward currents. While uncovering mechanistic effects on electrophysiology is interesting, the paper lacks some key details and contexts for non-expert readers to understand the importance of the work, e.g.:

Major:

For a non-expert in PICs, figure 1 is very difficult to follow. What are the orange and blue dashed lines? Where are the scale bars? What is a-PIC vs d-PIC? This first panel is an important setup for the rest of the paper, and the authors can do better at annotating and explaining PICs to make it easier for the reader. Panels B and C could also benefit from better labeling.

How the base model was developed and fit to experimental recordings is not well described and is missing from the results. Were there subtype models that fit to each subtype of PIC/spiking? How well does it capture the data? The authors jump to describing what happens when you play around with the PIC parameters without even convincing the reader that the model is a sufficient model to begin with. Moving figure 10 to be presented earlier in the paper results could help clarify these aspects.

Related to the above, why set NaP/Ca1.3 conductances to zero when manipulating the other parameter? It is unclear if the model still captures the experimental data it was fit to when doing this, and so it is unclear if manipulations on top of this will generate reliable results.

Can the authors demonstrate that the investigated ranges of conductance and kinetics in the Na-PIC and Ca-PIC models are within physiological ranges for these channels? The changes in dendritic length appear to be particularly large (so not surprising that the effects were dramatic).

Why is clockwise vs counterclockwise hysteresis important? The impact that it might have on spiking is not described. Also, why is it important in the context of locomotion?

Is the total Ca-PIC conductance kept fixed when increasing dendritic length? Otherwise, effects of distribution are conflated with effects due to increased total conductance.

Much of the discussion could be sharpened – a lot of it just re-states the results without adding enough contextualization of the results.

Minor:

Line 88: typo – “unknow”.

Figure 1B: What do the red vs blue dots mean?

Figure 1C: Panel A shows 6 types of PICs, but the proportions in this panel are only shown for three based on only the ∆V criteria. Probably to hint at overlap with firing type (∆V<0 vs ∆I<0). If the PIC and firing rate experiments were done on the same neurons, what are the actual overlaps between PIC types and firing types?

Lines 157-158 – do the authors mean to say “… soma, dendrites, axon initial segment, and axon”?

Line 283: type – “dominate” instead of “dominant”

Reviewer #2: **Summary **

The manuscript investigates the role of persistent inward currents (PICs), such as sodium and calcium channel currents, in serotonergic neurons of the medulla. It combines experimental and modeling techniques to explore the role of these currents in the different firing activities of these neurons. The experimental approach revealed different patterns of PICs and their associated neuronal firing activities. The computational approach helps modulate the currents by adjusting conductance and kinetics, specifically by tuning half-activation and ion channel gating.

**Global comments**

I believe that the paper presents interesting findings. However, the authors need to make major revisions:

I.The authors use specialized vocabulary that assumes prior knowledge, such as "hysteresis" (mentioned in the abstract and text), terms like "counterclockwise," and phrases such as "manifold types of repetitive firing" (in the title and text). The authors need to improve how they convey their message to a broad audience by ensuring that each concept is clearly defined or well-introduced. I strongly recommend improving this aspect of the manuscript. The introduction should provide clear definitions of important concepts. Additionally, the title and abstract should be reformulated to make them more accessible and engaging to a wider audience.

II.In contrast, some sections of the results (especially those related to Figures 8 and 9) could be summarized more concisely. The authors often describe figures in great detail, but there is a lack of perspective linking the computational experiments to biological neuronal behaviors. This often leads to the question, “And so what?” The authors must ensure that, whenever a computational experiment is conducted, the underlying logic behind it is clearly explained. Additionally, the implications of the results should be clearly conveyed and discussed.

III.I do not understand the choice of the terms “Ca-PIC” and “Na-PIC.” These currents seem to have already been described. If Ca-PIC represents the same thing as CaV1.3/CaL (mentioned on line 273, 339, 675), using different terms makes the message less clear. At the end of the text, the authors attempt to link them. If they are the same, the authors should use the most commonly accepted conventions, such as CaV1.3/CaL or NaP, and briefly describe how they work.

IV.The flow of the text in English needs to be improved to enhance the quality of the manuscript. I suggest consulting a scientific writing expert to help improve the overall structure and phrasing of some sentences. There are many repetitions, and the text length could be significantly reduced.

V.I also have a general comment about the figures. The quality of all figures needs to be revised. They should be presented in vector format to ensure clarity. Many figures are missing axis labels, color bar labels, and x-scale information.

VI.The authors use NEURON. The code containing the differential equations, as well as the code to reproduce the figures, must be made available on GitHub or a similar repository.

I found a number of issues that should be addressed before publication. Please see the complete report on the following pages. If any comments are expressed as questions, you should modify the text to make it clear on the first reading, so readers do not have to search for answers or look for missing explanations about a concept.

** Precise comments **

*Title

1. Line 1: The title needs to be changed. It does not convey a clear message. "Manifold types" is not a commonly used term in neuroscience. Similarly, in the abstract, "manifold types of repetitive firings" is not a standard term. If this phrase is used in the abstract, it should be defined.

Abstract

2. Line 21: Clarify the phrase “role in the locomotor system.”

3. Line 29: The phrase “P3-6” does not communicate effectively to the computational neuroscience community.

4. Line 32: It is unclear what is meant by “patterns of PIC and repetitive firings.

5. Line 34: This is presented as the second result of the paper, but in the results section, the Na-PIC analysis is discussed first. This is inconsistent.

6. Line 35: Avoid abbreviations in the abstract. “Vs” is not a standard term.

7. Line 39: The phrase “counterclockwise to clockwise” is not common or understandable to a broad audience. It requires reading the results section and referring to Figure 1 to comprehend the concept.

8. Line 53: Refer to comment 7 and the general comment. This concept should be clearly introduced and explained.

* Introduction

9. Line 80: The phrase “manifolds types of repetitive firings” is not a widespread concept in neuroscience. It even gives the impression to be AI-generated, as it does not show up in Google searches as an existing phrase.

10. Line 81: What does “underlying the generation of staircase PICs” mean? Please clarify.

11. Lines 94-95-98: Refer to comment 7. The concept is mentioned three times in four lines but remains very unclear. In the revised manuscript, this must be clearly addressed.

* Results

12. Line 139: Figure 1:

The quality of the figure is very poor. Everything is pixelated, even when downloaded in .tif format. There are no units or labels. Clear vertical and horizontal scales must be added, as well as appropriate labels. The traces should be in SVG, EPS, or PDF format for better quality.

13. Line 100-139: Overall Results Section for Figure 1:

The result section associated with this figure needs to be modified. Here is a potential suggestion:

(i) What is a PIC?

(ii) How are PICs categorized experimentally? Explain the method and terminology used.

(iii) Include a table or summary of the six patterns with the different values taken by the key categorizing parameters.

The overall flow and concepts are not enough clearly defined while the text relies on these definitions.

14. Line 105: There is no explanation of what a-PIC and d-PIC, Vonset, ∆V, etc., are. The definitions of the six patterns need to be improved. The diagram shown in Figure 10B should appear in Figure 1, along with Figure 10C to explain what ∆V and ∆I are. Without this, the text is unclear, and readers cannot easily distinguish between the patterns.

15. Line 116: A reference to the different traditional methods for classification is missing.

16. Line 153: This seems like a conclusion or a transition to the next paragraph. It seems like the title should appear earlier in the flow of the text.

17. Line 158: “Conductances” should be plural.

18. Line 159: Add a link to the methods section.

19. Line 165: The term “staircase PIC” is unclear and not defined or introduced properly.

20. Line 178: Why is gNa-PIC not tested below 100%, like at 50%? Why does it go up to 500%? The axis does not need to go to 600% if the data stops at 500%. Why not use absolute numbers (1, 2, 3, etc.) instead of percentages?

21. Line 208: For the blockage of Ca-PIC, specify that you set the conductance to zero.

22. Lines 218-219: You should refer to a figure or explain the concept.

23. Line 221: For subpanel 3C, consistency is needed:

• Either make the line and the label 'Vs=3mV' in the same color.

• Or make the line in the associated color but keep the label in black. The current mix of color codes between Vs=2 and 7 is confusing.

24. Line 221: Figure 3C is essentially your equation from line 698. It would make more sense to have it in panel A, as it involves adjusting Vs in the computational experiment. Then, show panels A and B.

25. Line 221: In Figure 3A, use a lowercase 's' for seconds: “1 s” instead of “1 S.”

26. Line 221: I would also suggest that in Figures 3D, 4C, and 5C, the horizontal axis be placed at the bottom of the panel, not at the top. This will make the graphs more consistent with the other three panels.

27. Line 221: Consistency is needed in the labels. In Figure 3, you fully describe 'm activation,' but in Figure 4, you just say 's_gate.'

28. Line 222: Clarify whether “recorded” refers to data simulated by the code or recorded in an experiment. This should be made clear immediately. I suggest reordering the figure subpanels for clarity.

29. Line 240: Indicate the method again.

30. Lines 250-251: This section describes the figure but does not explain the implications for neuronal activity.

31. Line 252: Replace the phrase “it’s worth noting” with something more formal, as this expression is not appropriate for a scientific article.

32. Line 275: Maintain consistent grammar; avoid mixing past and present tenses.

33. Line 280: Indicate how gNa is set to 0 (gNa=0).

34. Line 282: How did you decide on the range of changes for gCa-PIC? Why is it different from gNa-PIC?

35. Line 312: Why is Vs modulated to adjust the kinetics instead of tau? A better justification for the interest and necessity of adjusting the kinetics is missing.

36. Line 339: You mention that Cav3.1 is distributed along the dendrite, then continue by discussing Ca-PIC. The connection or distinction between Ca-PIC, Cav, and CaL is not clear. The introduction/motivation for investigating the distribution of Ca-PIC is insufficient.

37. Line 378: Are you blocking gCa as before?

38. Line 387: This sentence seems to restate the idea from line 378 with different words but the same structure. It is unclear why this repetition is necessary.

39. Line 389: What are the biological implications? Modify the text to explain the motivation and implications.

40. Line 393: In Figure 8, the squares in the maps do not help to visualize the mesh within the parameter space. A more accurate mesh should be used with more points.

41. Line 393: In Figure 8, the color bar label is missing.

42. Line 406: What are the implications for the neuron and the network? Modify the text to explain the motivation and implications.

43. Line 408: This sentence is almost identical to line 399. Should this be Ionset?

44. Line 421: Figure 9 needs improvement—the choice of axis and the color bar make the image hard to read and understand.

• In A1, hyperpolarization of V1/2 means moving up on the vertical axis but decreasing on the color bar axis.

• In B2, a reduction of Ioffset is blue, whereas in A3, a reduction of ∆V is yellow, similar to B3.

45. Lines 399-419: Consider condensing the text. For example:

• A depolarization of V1/2 is biologically represented/caused by...

• A reduction of Vs represents...

• A dendritic extension is represented/caused by...

These three modifications cause several effects:

• A1. Depolarization of Vonset, which implies...

• B1. Increase of Ionset, which implies... etc.

It also needs to be clearer which parameters are maintained, since some channels were blocked in previous experiments. We need to know if these conditions are still in effect or if they have returned to baseline.

* Discussion

46. Line 429: The concept of switching from clockwise to counterclockwise must be fully explained in the introduction and briefly reintroduced in the discussion. It is mentioned frequently but is not a standard term, and its implications or utility are not clear. This must be clarified in the revised manuscript.

47. Line 441: The title needs to be clearer. The next one, “Contributions of Ca-PIC to PIC patterns and firing types,” seems almost the same but more general.

48. Line 442: There are typos or strange symbols.

49. Line 442: Here, you state that L-type calcium (Ca-PIC) channels are mainly distributed on the dendrites of spinal neurons, but in line 339, you mention Cav3. You introduce Cav3 again in line 658. This needs to be made extremely clear in the revised manuscript. I believe that using the new notation Ca-PIC increases confusion, whereas using the appropriate scientific ion channel name makes it clearer.

50. Line 458: Why is the acronym defined again here? The same happens on line 464.

51. Line 491: The order of presentation in the discussion is inconsistent with the results. You first presented data on Na-PIC, then on Ca-PIC, but in the discussion, you reversed the order. You start by discussing Figure 7, then return to Figures 2 and 3.

Methods

52. Line 576: What is the content of the intracellular solution? If this is indicated in the following section, include a link.

53. Line 587: The phrase “line was tangent to the ascending current trace was defined as the onset of” is unclear. The mathematical description needs to be clearer.

54. Line 625: In Figure 10A, there is no scale bar to quantify the image. Acronyms might be added directly to the image to help understand the purpose of the slice image.

55. Line 625: Figures 10B and 10C are missing units.

56. Line 634: The code should be made available on GitHub or ModelDB.

57. Line 650: There is an error in the formatting of the title in the second column of Table 2.

58. Line 658: Do not introduce new conventions for acronyms.

59. Line 663: In "Vth," the “th” should be a subscript, like other variables (Ith, Em, Rin, etc.). This change should be applied consistently.

60. Line 667: Which parentheses are you referring to?

61. Line 672: The name should be consistent with the maximum conductance or vice versa.

62. Lines 675-676: Currents were introduced in line 652. Here, a different definition or vocabulary is used for the same concept. Consistency is needed throughout the manuscript.

**Have the authors made all data and (if applicable) computational code underlying the findings in their manuscript fully available?**

Reviewer #1: **No: ** The new computational code for this work was not made available. Albeit the model descriptions in the methods were sufficiently detailed, there are key details still missing in regards to how the models were fit or modified, as well as the manipulations in this work. I've outlined some of these questions in my comments, but having the code available would provide further clarification.

Reviewer #2: **No: ** The authors indicate in the “ Data Availability Statement.” The original contributions presented in the study are included in the article materials and methods, further inquiries can be directed to the corresponding author. I infer that the codes are not currently available. The model should be made accessible on a public repository, such as GitHub, or any other public platform. Additionally, the code needed to run the various computational experiments should also be provided.

PLOS authors have the option to publish the peer review history of their article (what does this mean? ). If published, this will include your full peer review and any attached files.

**Do you want your identity to be public for this peer review?** For information about this choice, including consent withdrawal, please see our Privacy Policy .

Reviewer #1: No

Reviewer #2: **Yes: ** Kathleen Jacquerie
---

## [Decision Letter · Decision Letter 1]

19 Dec 2024

PCOMPBIOL-D-24-00834R1

Multiple Patterns of Persistent Inward Currents with Multiple Types of Repetitive Firings in Medullary Serotonergic Neurons of Mice: An Experimental and Modeling Study

PLOS Computational Biology

Dear Dr. Dai,

Thank you for submitting your manuscript to PLOS Computational Biology. After careful consideration, we feel that it has merit but does not fully meet PLOS Computational Biology's publication criteria as it currently stands. Therefore, we invite you to submit a revised version of the manuscript that addresses the points raised during the review process.

Please submit your revised manuscript within 60 days Feb 18 2025 11:59PM. If you will need more time than this to complete your revisions, please reply to this message or contact the journal office at ploscompbiol@plos.org. Please include the following items when submitting your revised manuscript:

We look forward to receiving your revised manuscript.

Kind regards,

Peter Jedlicka

Guest Editor

PLOS Computational Biology

Lyle Graham

Section Editor

PLOS Computational Biology

**Additional Editor Comments:**

The authors have significantly improved to the manuscript, and the overall quality has greatly increased. However, there are still some key issues mentioned by one of the reviewers that need to be addressed.

**Reviewers' comments:**

Reviewer's Responses to Questions

**Comments to the Authors:**

Reviewer #1: I am satisfied with the changes made by the authors.

A few remaining minor comments:

-Line 133: The term “segment” in NEURON describes compartments. Probably “section lists” is more correct here.

-Lines 388-389: I suggest re-phrasing to something like this “When we increased the dendritic length, we also extended the CaL distribution to the new dendritic compartments.”

-Line 655: correction - “…became more dramatic with…”

Reviewer #2: The authors have done an excellent job reorganizing the figures and text, which now clearly highlights the quality of the work. However, there are still some major concerns that need to be addressed before the manuscript can be accepted for publication.

Global comment:

I.Definition of trajectory: In my initial review, I pointed out that the use of specialized jargon was making the manuscript difficult to understand. For instance, the term "trajectory" was used instead of "hysteresis," which raises concerns. In the abstract, the term "trajectory" is introduced without sufficient context, making it unclear. Specifically, it is important to clarify: the trajectory of which variable? In which plane? And what does this mean for the firing activity?

On line 82, the term "trajectory" appears for the first time: "If the a-PIC is larger than the d-PIC, the trajectory of the PIC folded from the midline appears counterclockwise; otherwise, a clockwise trajectory of PIC forms." However, this sentence does not clearly define what is meant by "trajectory." It lacks specificity about which trajectory is being referred to and what variables or axes are involved.

In Figure 2, the trajectory is still presented without labeled axes, units, or any clarifying information, while in Figure 3, the y-axis is current and the x-axis is time. Based on my understanding, the term "trajectory" seems to refer to the trajectory of the current in the current-voltage plane, showing how the current evolves. You could phrase it as: "The trajectory of the PIC current shows the evolution of the current in the current-voltage plane. This trajectory can be counterclockwise if a-PIC is larger than d-PIC..." [continue with the rest of line 83].

The repeated issue of using the term "trajectory" in the abstract without context, not defining it in the introduction, and leaving figures without proper labeling indicates a lack of clarity that must be addressed before publication. Since "trajectory" is a key concept used throughout the manuscript, it needs to be properly defined and consistently applied to ensure the paper is understandable to a broader audience.

II.Quantification of the firing activity: The revised manuscript has significantly improved over the initial version. I appreciated the description of the firing patterns using the parameters Ion, Ioff, Von, and Voff. While these indicate changes in the IV curve, they do not sufficiently explain their connections or their effects on real-world cellular behavior. After this second review, I would like to suggest adding one additional analysis. It seems necessary to quantify the properties of the firing neuron. For instance, I suggest including measurements of the maximum firing frequency under different conditions (gNaP ranging from 1x to 6x, gCaL, and CaL length), the duration of the spiking, and the onset time before spiking begins. Specifically, the maximum frequency could be extracted directly from the peak of the f-I curve, and the duration of the spiking activity could be defined as the length of the plateau phase. These parameters would provide a clearer connection to what you introduce regarding "various physiological processes in neurons, including the generation and modulation of action potentials, amplification of synaptic current, and regulation of neuronal excitability."

It is striking in Figures 3B, 6B, 7B, and 8B that firing frequency is affected by changes in parameters such as gCaL, gNaP, and gCaL distance. For example, the distribution length of gCaL drastically alters the duration of the spiking activity. These parameters likely impact network activity and should be quantified. The fact that Vs can alter spiking duration, frequency, and initiation time could have significant effects on network behavior (as shown in Figure 4). Quantifying these aspects would strengthen the connection between the presented data and the physiological relevance of these findings.

III.Variability in the conductance model: Have you considered introducing variability in other conductances, as a real neuron would naturally encounter? While the results are compelling, they may only be accurate for the specific set of parameters chosen. Incorporating variability in additional conductances could provide a more realistic representation of neuronal behavior and strengthen the robustness of your findings. You can replicate your results obtained on this particular set of parameters with a small variability added to your intrinsic parameters (other conductances) and you will compute the mean and the standard deviation of your characteristics (Ion, Ioff, …). It will make the message of Figures 3 to 8 more robust.

Figures in general

- Figure 3: Instead of only showing the trajectory, I recommend replicating Figure 2A and displaying the current trace with the voltage bi-ramp, placing the trajectory in a subpanel to the right of the traces. Similarly, for Figure 3B, I suggest showing the actual firing activity of the neuron model at 100% and 300% of gNaP. I understand your rationale for using percentages to quantify conductance, but indicating 1x and 3x would also be effective. You have added axes everywhere. it is fantastic. But you forgot to add a reference. Add the resting membrane voltage for your voltage traces (example in figure 4C, 1C) and the current same. It is nice to have a scale bar but for example, we also need to see the 0Hz start point on figure 3B.

- For Figure 8, could you add a plot showing the change in frequency as a function of length, similar to what you did in Figure 7B?Line 536: which f-I curve? example figure 3-B, 6b, you indicate f-seconds. Could you double check all your axes.

Here is the list of typos or small modifications to bring to the text:

* Abstract *

• Line 26: Change "Ionic bases" to "ionic mechanisms."

• Line 30-32: Suggested rewording: "PICs in 5-HT neurons were classified into six patterns based on their response to the applied voltage ramp. Three firing activities were defined according to their response to the current ramp injection."

• Line 41: For clarity, consider: "This study provides insights into the ionic mechanisms underlying the generation of multiple PIC patterns..."

* Introduction *

Consider emphasizing this result in the abstract, as it is quite significant. If it has already been included, it may benefit from clearer phrasing to ensure the importance of the finding is effectively conveyed

* Results *

• Line 136: In Table 1, subscript 'm' in Eₘ and in Rₗₙ.

• Line 176: Add axis labels for the trajectory—current on the y-axis (I) and voltage on the x-axis (V). This is still unclear.

• Line 183: I like the horizontal dotted lines for Ion and Ioff. Add them to Panel A for Von and Voff as well.

• Line 232: The x-axis in the trajectory plot is incorrect. The trajectory is the line on the I-V plane.

• Line 234: In Figures 3A-B, add the dot and triangle from Figure 3C to directly show Von, Voff, Ion, and Ioff.Line 234: Could you add on Fig3A-B the dot and the triangle associated with Fig3C. We can directly see the Von, Voff, Ion and Ioff.

• Line 234: In Figures 3A-B, add the dot and triangle from Figure 3C to directly show Von, Voff, Ion, and Ioff.

• Line 240: In Figure 3F, you describe a slight change with a slope of 0.00002, but in Figure 4G, you say no change with a slope of 0.007. These descriptions are inconsistent.

• Line 264 (Figure 4): Follow a consistent convention for the y-axis labels (m_{in,NaP} ) in the equation and throughout the text. In Panel A, there are two different conventions. Add "( m_inf,NaP)" in the legend.

• Line 266: Use "bi-ramp" or "ramp" consistently throughout the paper.

• Line 270: Color-code the triangle and circle in red and orange to match Panel A.

* Discussion *

• Line 596: The term "hysteresis" is still present.

• Line 603: Consider moving this section to the introduction. If this is not your result, it would help readers understand why you chose "trajectory" as a key factor for analysis throughout the paper.

• Line 609, 611, 618: Avoid repeating "in this present study" within such close proximity.

• Line 613: Include the frequency, duration, and spiking initiation time in the discussion (refer to general comment II).

* Methods*

• Line 787: Consider adding the full name of the channel each time. For example, in line 786 you use "m," and again in line 802. It’s standard in computational modeling papers to follow the convention of using activation variables followed by the ion channel notation. The same applies m_inf,NaP … etc. This was mentioned in the previous review as well: "The same notation 'm' is used as in line 681. I suggest adding subscripts to differentiate the currents described, such m_Na, h_Na, and m_NaP

• I would like to emphasize that the codes and the models are available in the github.

**Have the authors made all data and (if applicable) computational code underlying the findings in their manuscript fully available?**

Reviewer #1: Yes

Reviewer #2: Yes

PLOS authors have the option to publish the peer review history of their article (what does this mean? ). If published, this will include your full peer review and any attached files.

**Do you want your identity to be public for this peer review?** For information about this choice, including consent withdrawal, please see our Privacy Policy .

Reviewer #1: No

Reviewer #2: **Yes: ** Kathleen Jacquerie

**Figure resubmission:**
---

## [Decision Letter · Decision Letter 2]

7 Feb 2025

PCOMPBIOL-D-24-00834R2

Multiple Patterns of Persistent Inward Currents with Multiple Types of Repetitive Firings in Medullary Serotonergic Neurons of Mice: An Experimental and Modeling Study

PLOS Computational Biology

Dear Dr. Dai,

Thank you for submitting your manuscript to PLOS Computational Biology. After careful consideration, we feel that it has merit but does not fully meet PLOS Computational Biology's publication criteria as it currently stands. Therefore, we invite you to submit a revised version of the manuscript that addresses the points raised during the review process.

Please submit your revised manuscript within 30 days Apr 09 2025 11:59PM. If you will need more time than this to complete your revisions, please reply to this message or contact the journal office at ploscompbiol@plos.org. Please include the following items when submitting your revised manuscript:

We look forward to receiving your revised manuscript.

Kind regards,

Peter Jedlicka

Guest Editor

PLOS Computational Biology

Lyle Graham

Section Editor

PLOS Computational Biology

**Reviewers' comments:**

Reviewer's Responses to Questions

**Comments to the Authors:**

Reviewer #2: Thank you for the modifications to the paper. The revised version is now clearer than the previous one.

I understand that you want to stand by your version of the manuscript without conducting additional computational experiments, and that you wish to save them for a future paper. This would have added significant value to the current paper and strengthened the message.

I still notice some recurring inconsistencies with the figure and trajectory labels. We are converging toward a polished version, but wherever you show a trajectory, the I-V axis is needed.

Minor Review

- Figure 2A: The trajectory I vs V is still missing an axis. There are V vs time and I vs time plots, but the trajectory is never associated with the I-V axis.

- Figure 3: The axes are incorrect. According to your axes, the current can move backward in time (to the left). Once again, a trajectory is defined in I-V space. Time is the variable that is used to move along the axis (indicated by the arrow). The bi-ramp consists of the increase and decrease of V, which is why you cannot align a trajectory with just half of the stimulus. It should be the same as in Figure 2, where the stimulus is shown in full.

- Figure 6: There is the same inconsistency with the axes. The current cannot move backward in time. This should be represented in I-V space, with the arrow indicating the progression along the time axis.

- Methods: Thank you for modifying the notation for the m and h variables in the conductance-based model. However, there is still an underscore between the m or h and the subscript of the current name (e.g., m_NA). The underscore must be removed and the subscript must stay. This formatting should be updated for each current.

**Have the authors made all data and (if applicable) computational code underlying the findings in their manuscript fully available?**

Reviewer #2: Yes

PLOS authors have the option to publish the peer review history of their article (what does this mean? ). If published, this will include your full peer review and any attached files.

**Do you want your identity to be public for this peer review?** For information about this choice, including consent withdrawal, please see our Privacy Policy .

Reviewer #2: **Yes: ** Kathleen Jacquerie

**Figure resubmission:**
---

## [Editor Report · Decision Letter 3]

26 Feb 2025

Dear Professor Dai,

We are pleased to inform you that your manuscript 'Multiple Patterns of Persistent Inward Currents with Multiple Types of Repetitive Firings in Medullary Serotonergic Neurons of Mice: An Experimental and Modeling Study' has been provisionally accepted for publication in PLOS Computational Biology.

Best regards,

Peter Jedlicka

Guest Editor

PLOS Computational Biology

Lyle Graham

Section Editor

PLOS Computational Biology

---

## [Editor Report · Acceptance letter]

PCOMPBIOL-D-24-00834R3

Multiple Patterns of Persistent Inward Currents with Multiple Types of Repetitive Firings in Medullary Serotonergic Neurons of Mice: An Experimental and Modeling Study

Dear Dr Dai,

I am pleased to inform you that your manuscript has been formally accepted for publication in PLOS Computational Biology. Your manuscript is now with our production department and you will be notified of the publication date in due course.

With kind regards,

Anita Estes
